# PRETRAINING A SHARED Q-NETWORK FOR DATA-EFFICIENT OFFLINE REINFORCEMENT LEARNING

## ABSTRACT

Offline reinforcement learning (RL) aims to learn a policy from a static dataset without further interactions with the environment. Collecting sufficiently large datasets for offline RL is exhausting since this data collection requires colossus interactions with environments and becomes tricky when the interaction with the environment is restricted. Hence, how an agent learns the best policy with a minimal static dataset is a crucial issue in offline RL, similar to the sample efficiency problem in online RL. In this paper, we propose a simple yet effective plug-and-play pretraining method to initialize a feature of a $Q$-network to enhance data efficiency in offline RL. Specifically, we introduce a shared $Q$-network structure that outputs predictions of the next state and $Q$-value. We pretrain the shared $Q$-network through a supervised regression task that predicts a next state and trains the shared $Q$-network using diverse offline RL methods. Through extensive experiments, we empirically demonstrate that the proposed method enhances the performance of existing popular offline RL methods on the D4RL and Robomimic benchmarks, with an average improvement of 135.94% on the D4RL benchmark. Furthermore, we show that the proposed method significantly boosts data-efficient offline RL across various data qualities and data distributions. Notably, our method adapted with only 10% of the dataset outperforms standard algorithms even with full datasets.

## 1 INTRODUCTION

Sample efficiency is a crucial issue in reinforcement learning (RL) since typical RL considers an online learning nature that involves iterative processes between experience collections and policy improvements through online interactions with the environment (Sutton et al., 1998). Unfortunately, requiring excessive online interactions is impractical in several cases since data collection requires expensive costs and retains potential risks of the agent, e.g. hardware corruption. Offline RL is one approach to alleviate this sample efficiency problem, which provides a solution by avoiding online interactions with the environment (Levine et al., 2020). In recent years, pretraining with offline RL and fine-tuning with online RL have been investigated to improve sample efficiency of the online interactions (Nakamoto et al., 2024; Xie et al., 2021; Rafailov et al., 2023; Ball et al., 2023).

Similar to addressing the sample efficiency problem in online RL, learning offline RL with minimal datasets is necessary since collecting enormous experience charges expensive costs and unfavorable explorations, hampering the possibility of offline RL in the real world. In this paper, we name this problem as data efficiency where an agent tries to learn the best policy with minimal data in the offline RL scheme. Despite the necessity of data efficiency, this problem has not been treated enough in previous works. Although some researchers have evaluated their work empirically on reduced datasets in part of the experiments (Agarwal et al., 2020; Kumar et al., 2020a;b), they have overlooked this data efficiency problem. In the case of online RL, model-based RL and representation learning have proposed the resolution of sample efficiency problem (Sutton, 1991; Hafner et al., 2019b; Schwarzer et al., 2020; 2021). As in online RL, one can expect that offline model-based RL or representation method might resolve this data efficiency problem (Yu et al., 2020; Sun et al., 2023; Yang & Nachum, 2021). However, Figure 7 demonstrates that both approaches are unable to overcome this problem.

In this work, we propose a simple yet effective plug-and-play method that pretrains a shared $Q$-network toward data-efficient offline RL. Specifically, the shared $Q$-network structure is composed of

Figure 1: **Overview of our pretraining method.** Our method splits the original $Q$-network into two core architectures: a shared network that extracts the representation $z$ from the concatenated vector of state $s$ and action $a$ and separated heads for training the transition model network and $Q$-network, respectively.

two parts as illustrated in Figure 1. First, a shared deep neural network layer ($h_\varphi$) takes the state and action pair as inputs. Second, separate shallow output parts ($g_\psi$ and $f_\theta$) consist of two linear layers that individually output a $Q$-value for a $Q$-function and a next state prediction for a transition model. The learning phase of the shared $Q$-network consists of a pretraining and an RL training phase. In the pretraining phase, the shared network attached with a shallow transition layer ($h_\varphi$ and $g_\psi$) is trained through a supervised regression task that predicts the transition model. After the pretraining phase where the shared network is initialized with the pretraining, the shared network is connected with a shallow $Q$ layer ($h_\varphi$ and $f_\theta$) and trained with an existing offline RL value learning.

We empirically demonstrate that our method improves the performance of existing popular offline RL methods on the D4RL (Fu et al., 2020), and Robomimic (Mandlekar et al., 2021), benchmarks with an average improvement of $135.94\%$ on the D4RL benchmark. We also show that our method maintains data-efficient performance with fragments of the dataset across the *data quality* on the D4RL dataset. Moreover, we investigate our method across the *data collection strategies* on the ExoRL datasets (Yarats et al., 2022), assuming a small dataset would have a shifted data distribution compared to a large dataset. As a result, we demonstrate that our method improves the performance regardless of the qualities of the datasets and the data distributions. Figure 6 and Figure 9 show that our method with 10% of datasets outperforms vanilla algorithms even with full datasets. Furthermore, Figure 7 demonstrates that our method indeed outperforms the offline model-based RL and representation approaches in reduced datasets.

## 2 RELATED WORKS

**Offline RL.** Offline RL aims to learn a policy with static data without further interactions with the environment. Previous approaches have mainly addressed the distribution shift problem, which is caused by the idea that queries of the $Q$-function over out-of-distribution actions may yield overly optimistic values during offline training (Fujimoto et al., 2019; Kumar et al., 2019; Levine et al., 2020; Kumar et al., 2020b; Fujimoto & Gu, 2021; Kostrikov et al., 2021a). Recently, scalability to a large dataset and neural network model has been studied (Chebotar et al., 2023; Padalkar et al., 2023; Team et al., 2024). In other fields, pretraining with offline RL and fine-tuning with online RL is examined to improve sample efficiency in the online interaction step (Nakamoto et al., 2024; Xie et al., 2021; Rafailov et al., 2023; **?**). In contrast, distinct experiments over the way to consuming the static dataset have been conducted, e.g., an imbalanced dataset, unlabeled data, and even data corruption under an offline RL scheme (Hong et al., 2023; Yu et al., 2022; Yang et al., 2023). While prior research (Agarwal et al., 2020; Kumar et al., 2020a;b) often has evaluated their work on reduced datasets as a partial result, the field overlooks the data efficiency problem itself as a main contribution. In contrast, we aim to improve the data efficiency in offline RL (i.e., learning the best policy with minimal data). In this work, we propose a simple yet effective plug-and-play method for pretraining a shared $Q$-network toward the data-efficient offline RL.

**Sample efficient RL.** A common issue in most RL algorithms is sample efficiency: excessive interactions with the environment are required to learn an optimal policy. For this reason, sample efficiency has been an active research topic in RL (Kostrikov et al., 2021b; Yarats et al., 2021c;

D'Oro et al., 2022). Model-based RL (Sutton, 1991; Deisenroth & Rasmussen, 2011; Hafner et al., 2019b;a; Hansen et al., 2022) is a common approach to resolve sample inefficiency by learning a (latent) dynamics model and using it to generate additional transition samples. Otherwise, effective pretraining (Schwarzer et al., 2021; Yarats et al., 2021c) and data augmentation (Laskin et al., 2020; Kostrikov et al., 2021b) play a critical role in improving sample efficiency in RL. Recently, offline-to-online (Lee et al., 2022; Ball et al., 2023; Rafailov et al., 2023; Feng et al., 2023; Nakamoto et al., 2024) and foundation model (Ahn et al., 2022; Seo et al., 2022; Brohan et al., 2023b;a; Bhateja et al., 2023) have tackled this problem where the poor sample efficiency of online RL regime is alleviated by leveraging large offline data. In this paper, we separately define the data efficiency problem in offline RL as the ability of an offline RL algorithm how an agent can learn the best policy even with minimal pre-collected samples called dataset in offline RL. We claim that this data efficiency problem is different from the sample efficiency problem since online RL has opportunities for interactions with environments which can present another chance to improve the sample efficiency.

## 3 MARKOV DECISION PROCESS

We consider the Markov decision process, where the agent sequentially takes actions to maximize cumulative discounted rewards. In a Markov decision process with the state-space $\mathcal{S} := \{1, 2, \ldots, |\mathcal{S}|\}$ and action-space $\mathcal{A} := \{1, 2, \ldots, |\mathcal{A}|\}$, the decision maker selects an action $a \in \mathcal{A}$ at the current state $s \in \mathcal{S}$, then the state transits to the next state $s' \in \mathcal{S}$ with probability $P(s'|s, a)$, and the transition incurs a reward $r(s, a, s') \in \mathbb{R}$, where $P(s'|s, a)$ is the state transition probability from the current state $s \in \mathcal{S}$ to the next state $s' \in \mathcal{S}$ under action $a \in \mathcal{A}$, and $r(s, a, s')$ is the reward function. For convenience, we consider a deterministic reward function and simply write $r(s_k, a_k, s_{k+1}) =: r_k, k \in \{0, 1, \ldots\}$.

A deterministic policy, $\pi : \mathcal{S} \to \mathcal{A}$, maps a state $s \in \mathcal{S}$ to an action $\pi(s) \in \mathcal{A}$. The objective of the Markov decision problem is to find a deterministic (or stochastic) optimal policy, $\pi^*$, such that the cumulative discounted rewards over infinite time horizons is maximized, i.e.,

$$\pi^* := \arg\max_{\pi \in \Theta} \mathbb{E}\left[\sum_{k=0}^{\infty} \gamma^k r_k \,\middle|\, \pi\right],$$

where $\gamma \in [0, 1)$ is the discount factor, $\Theta$ is the set of all deterministic policies, $(s_0, a_0, s_1, a_1, \ldots)$ is a state-action trajectory generated by the Markov chain under policy $\pi$, and $\mathbb{E}[\cdot|\pi]$ is an expectation conditioned on the policy $\pi$. Moreover, Q-function under policy $\pi$ is defined as

$$Q^\pi(s, a) = \mathbb{E}\left[\sum_{k=0}^{\infty} \gamma^k r_k \,\middle|\, s_0 = s, a_0 = a, \pi\right], \quad (s, a) \in \mathcal{S} \times \mathcal{A}.$$

## 4 PRETRAINING Q-NETWORK WITH TRANSITION MODEL HELPS IMPROVING DATA EFFICIENCY

In this paper, we propose a simple yet effective pretraining method adapting features of the transition model into the initialization of $Q$-network to improve data efficiency in offline RL. To this end, we design $Q$-network that partially shares a network with the estimation of the transition model. In particular, the transition model is constructed as follows:

$$\hat{s}' = (g_\psi \circ h_\varphi)(s, a), \quad (s, a) \in \mathcal{S} \times \mathcal{A}, \tag{1}$$

where $\hat{s}'$ is the estimated next state, $g_\psi$ is a parameterized linear function, and $h_\varphi$ is shared with the $Q$-network, which is defined as

$$Q_{\varphi,\theta}(s, a) = (f_\theta \circ h_\varphi)(s, a), \quad (s, a) \in \mathcal{S} \times \mathcal{A}, \tag{2}$$

where $f_\theta$ is also a parameterized linear function that represents the linear output layer and $h_\varphi$ represents the fully connected neural network layers shared with the transition model in (1). The overall structures of the neural networks are illustrated in Figure 1.

In the proposed method, the transition model $g_\psi \circ h_\varphi$ is pretrained by minimizing the mean squared prediction error loss function

$$\mathcal{L}_{pre}(\varphi, \psi) = \sum_{(s,a,s')\in\mathcal{D}} (s' - (g_\psi \circ h_\varphi)(s, a))^2 \tag{3}$$

---

**Algorithm 1** Pretraining Q-network scheme for Offline RL

---

**Input**: Dataset $\mathcal{D}$ of transition $(s, a, s')$, learning rate $\alpha$
Initialize parameters $\varphi, \psi$
**for** each gradient step **do**
 Sample a mini-batch $\mathcal{B} \sim \mathcal{D}$
 Compute the transition model estimation error

$$\mathcal{L}_{pre}(\varphi, \psi) = \sum_{(s,a,s') \in \mathcal{B}} (s' - (g_\psi \circ h_\varphi)(s, a))^2$$

 Update weights of the shared network and transition model network

$$\varphi \leftarrow \varphi - \alpha \nabla_\varphi \mathcal{L}_{pre}(\varphi, \psi), \quad \psi \leftarrow \psi - \alpha \nabla_\psi \mathcal{L}_{pre}(\varphi, \psi)$$

**end for**
**Output**: Pretrained weights $\varphi$ of the shared network

---

over the pre-collected dataset $\mathcal{D}$ which includes a given set of the transition $(s, a, s')$. Afterward, the pretrained parameter $\varphi$ can be used as an initial or fixed parameter for standard RL algorithms based on the $Q$-network structure in (4) without any modification. The overall pretraining process is summarized in Algorithm 1 for offline RL. We also note that similar principles can be applied for online RL as well, and the corresponding algorithm is given in Appendix A.

Later in this paper, we empirically demonstrate that combining the proposed pretraining method with existing offline RL methods can effectively improve their performances. Moreover, we demonstrate that our method indeed improves data efficiency through some experiment settings in offline RL.

### 4.1 ANALYSIS: BASED ON THE PROJECTED BELLMAN EQUATION

In this section, we analyze how our method can resolve the data efficiency problem from the perspective of the projected Bellman equation. For simplicity and convenience of presentation, we assume that the state and action spaces are discrete and finite, and the transition is deterministic. However, the principles in this paper can be extended to more general continuous state and continuous action cases. Our analysis is based on the observation that $Q$-function with neural networks can be generally represented by (2). Defining the feature vector $z = h_\varphi(s, a) \in \mathbb{R}^m$, it can be rewritten as

$$Q_{\varphi,\theta}(s, a) = \sum_{i=1}^{m} \theta_i h_{\varphi,i}(s, a) = \langle \theta, h_\varphi(s, a) \rangle, \quad (s, a) \in \mathcal{S} \times \mathcal{A}. \tag{4}$$

When $\varphi$ is fixed, then the above structure can be viewed as a linear function approximation with the feature function $h_{\varphi,i}$. In the proposed method, $h_{\varphi,i}$ is indeed pretrained by minimizing the loss in (3) and then fixed while learning $Q$-function in (4). Therefore, the interpretation based on the linear function approximation is expected to be a reasonable model to explain the phenomenon in the proposed method.

It is well known that with linear function approximation, the corresponding standard Bellman equation

$$Q_{\varphi,\theta}(s, a) = R(s, a) + \gamma \sum_{s' \in S} P^\pi(s'|s, a) \sum_{a' \in A} Q_{\varphi,\theta}(s', a')$$

may not admit a solution in general. However, typical TD-learning algorithms are known to converge to the unique fixed point of the projected Bellman equation. In particular, considering the vector form of the Bellman equation, $Q_{\varphi,\theta} = R + \gamma P^\pi Q_{\varphi,\theta}$, the projected Bellman equation (Melo & Ribeiro (2007)) is known to admit a solution

$$Q_{\varphi,\theta} = \Pi(R + \gamma P^\pi Q_{\varphi,\theta})$$

where $\Pi$ is the projection onto the column space, $C(H_\varphi)$, of the feature matrix $H_\varphi$ defined as

$$H_\varphi := \begin{bmatrix} \vdots \\ h_\varphi(s, a)^T \\ \vdots \end{bmatrix}.$$

Table 1: **The Rank of the latent space of Q-network on the D4RL benchmark.** We compare the rank of the latent space between a vanilla TD3+BC and TD3+BC adapted with our method over 512 samples. As a result, adapting our method significantly increases the rank of the latent space, leading to reduced approximation error.

| | Halfcheetah | | Hopper | | Walker2d | |
| | TD3+BC | TD3+BC (+ours) | TD3+BC | TD3+BC (+ours) | TD3+BC | TD3+BC (+ours) |
|---|---|---|---|---|---|---|
| Random | 59 | **236** | 69 | **192** | 72 | **82** |
| Medium | 55 | **249** | 85 | **227** | 55 | **254** |
| Medium Replay | 49 | **252** | 77 | **249** | 77 | **255** |
| Medium Expert | 58 | **236** | 86 | **232** | 52 | **253** |
| Expert | 44 | **198** | 104 | **198** | 68 | **225** |

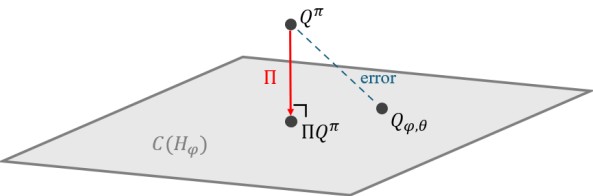

Figure 2: **Reduced approximation error with the expanded column space of $H_\varphi$.** In linear approximation, there exists $Q^\pi$ outside of the column space of $H_\varphi$. To deal with this problem, the projected Bellman equation projects $Q^\pi$ to $\Pi Q^\pi$ which exists in the column space of $H_\varphi$.

The corresponding solution is known to have the error bound

$$||Q_{\varphi,\theta} - Q^\pi||_\infty \leq \frac{1}{1-\gamma}||\Pi Q^\pi - Q^\pi||_\infty, \tag{5}$$

where $Q^\pi$ is the true $Q$-function corresponding to the target policy $\pi$. As can be seen from the above bound, the error depends on the feature matrix $H_\varphi$. We can observe that the smaller the distance between $C(H_\varphi)$ and $Q^\pi$, the smaller the error between $Q_{\varphi,\theta}$ and $Q^\pi$. Therefore, a proper choice of the feature function is key to the successful estimation of $Q^\pi$.

With the neural network function approximation, typical value-based RL algorithms update both $\varphi$ and $\theta$ simultaneously via TD-learning algorithms. Since the feature functions, $h_{\varphi,i}$, are in general nonlinear and non-convex in $\varphi$, it may sometimes converge to a local optimal solution. This in turn implies that appropriate initialization or pretraining of the feature functions, $h_{\varphi,i}$, can play an important role for estimating $Q$-function with smaller approximation errors on the right-hand side of (5) by avoiding suboptimal local solutions.

We conjecture that the pretraining approach with the transition model introduced in the previous section can effectively shape the feature functions so that the column space $C(H_\varphi)$ can cover higher dimensional vector space in $\mathbb{R}^{|S \times A|}$. As shown in Figure 2, this eventually results in a reduction of the solution error on the right-hand side of (5). To support this, we empirically compare the rank of the Q-network in the latent space between vanilla and the pretrained TD3+BC with our method over 512 data samples.

Table 1 exhibits that adapting our method shows a significantly higher rank than the rank of the vanilla method. From the results, we claim that the proposed method indeed expands the column space $C(H_\varphi)$ and covers higher dimensional vector space in $\mathbb{R}^{|S \times A|}$, leading to more precise Q-function estimation. In other words, we might learn a more precise Q-function with the same amount of samples, and it means that we can get a desirably estimated Q-function with less data. In the following section, we demonstrate our claim with empirical experiments.

## 5 EXPERIMENTS

In this section, we evaluate our method over existing offline RL methods with the popular offline RL benchmarks, D4RL, and the more complex domain, Robomimic. Furthermore, we examine the proposed method over the partial fragments of D4RL and ExoRL datasets for data-efficient offline RL. We introduce a detailed experimental setup and baselines in the following paragraphs and provide empirical results subsequently.

**Experimental setup.** We have considered heterogeneous tasks and diverse datasets for precise comparisons. For the locomotion task, the proposed method is compared with existing methods in the popular D4RL benchmark (Fu et al., 2020). Three different embodied agents and five distinct datasets are considered in order to validate the effectiveness of the proposed method: *HalfCheetah, Hopper, Walker2d* for agents and *random, medium-replay, medium, medium-expert, expert* for datasets. For the tabletop manipulation tasks, we have evaluated the proposed method in the Robomimic benchmark, (Mandlekar et al., 2021), where off-the-shelf offline RL methods are already implemented. Two different tabletop tasks and mixed-quality datasets are considered to verify the scalability of the proposed method: *Lift, Can* for tasks and *Machine-Generated (MG)* for datasets. For data-efficient offline RL, we have evaluated the proposed method across the reward qualities of the datasets of D4RL *Gym locomotion tasks*, and the dataset collection strategies for *walker walk (i.e. SMM, RND, ICM)* and *point mass maze (i.e. Proto, Diayn)* in ExoRL (Yarats et al., 2022). See Appendix C for a more detailed setup for tasks and datasets.

**Baselines.** We have designed extensive experiments on the D4RL benchmark to verify the effectiveness of the proposed method built on top of the popular offline RL methods, including AWAC (Nair et al., 2020), CQL (Kumar et al., 2020b), TD3+BC (Fujimoto & Gu, 2021), and IQL (Kostrikov et al., 2021a). To verify the benefits of the proposed method, we compared the normalized scores between the vanilla method and the one combined with the proposed pretraining method. Similar to the D4RL benchmark, the success rate is compared on the Robomimic benchmark, where IQL, TD3+BC, BCQ (Fujimoto et al., 2019), and IRIS (Mandlekar et al., 2020), were used in combination with the proposed methods. We also evaluate MOPO (Yu et al., 2020), MOBILE (Sun et al., 2023) and ACL (Yang & Nachum, 2021) to compare the proposed method with offline model-based RL and representation approaches. On the ExoRL benchmark, we used TD3 (Fujimoto et al., 2018), for *walker walk* task, and CQL for *point mass maze* tasks. See Appendix E for more implementation details.

Table 2: **Averaged normalized scores on the D4RL benchmark over 5 seeds.** In each column corresponding to different RL methods, values on the left-hand side are scores of the baseline methods directly taken from the literature. The values on the right-hand side of each column represent scores of the proposed methods combined with the baselines. The increased scores compared to the baselines are highlighted in blue font, and they are reported with the mean and standard deviations over five random seeds.

| | | AWAC | CQL | IQL | TD3+BC |
|---|---|---|---|---|---|
| Random | HalfCheetah | 2.2→51.10±0.89 | 21.7±0.9→31.94±2.63 | →18.28±1.02 | 10.2±1.3→14.83±0.54 |
| | Hopper | 9.6→59.47±33.79 | 10.7±0.1→30.20±2.66 | →10.67±0.41 | 11.0±0.1→31.56±0.16 |
| | Walker2d | 5.1→13.11±3.91 | 2.7±1.2→19.56±4.49 | →8.88±0.71 | 1.4±1.6→11.23±5.05 |
| Medium | HalfCheetah | 37.4→54.63±1.45 | 37.2±0.3→39.93±18.84 | 47.4→48.85±0.16 | 42.8±0.3→49.17±0.26 |
| | Hopper | 72.0→101.73±0.20 | 44.2±10.8→90.58±2.23 | 66.4→78.62±2.21 | 99.5±1.0→71.52±2.16 |
| | Walker2d | 30.1→89.51±0.88 | 57.5±8.3→84.66±0.67 | 78.3→83.63±1.14 | 79.7±1.8→87.09±0.60 |
| Medium Replay | HalfCheetah | →55.75±1.30 | 41.9±1.1→47.60±0.37 | 44.2→45.48±0.17 | 43.3±0.5→45.84±0.26 |
| | Hopper | →106.67±0.59 | 28.6±0.9→98.63±2.12 | 94.7→99.43±1.71 | 31.4±3.0→100.16±1.60 |
| | Walker2d | →100.31±2.11 | 15.8±2.6→87.66±1.30 | 73.9→87.95±1.68 | 25.2±5.1→92.01±1.58 |
| Medium Expert | HalfCheetah | 36.8→90.05±1.89 | 27.1±3.9→82.75±6.51 | 86.7→95.25±0.14 | 97.9±4.4→96.89±0.92 |
| | Hopper | 80.9→113.23±0.22 | 111.4±1.2→111.06±0.81 | 91.5→105.77±11.31 | 112.2±0.2→113.02±0.19 |
| | Walker2d | 42.7→111.88±0.28 | 68.1±13.1→91.63±42.48 | 109.6→112.09±0.93 | 101.1±9.3→111.58±0.35 |
| Expert | HalfCheetah | 78.5→93.48±0.11 | 82.4±7.4→97.09±1.03 | →97.40±0.13 | 105.7±1.9→98.86±0.55 |
| | Hopper | 85.2→112.86±0.10 | 111.3±2.1→112.10±0.35 | →113.34±0.46 | 112.2±0.2→113.35±0.28 |
| | Walker2d | 57.0→111.22±0.35 | 103.8±7.6→110.64±0.28 | →112.80±1.08 | 105.7±2.7→111.00±0.15 |
| Total | | →1265.01±48.07 | 764.3±61.5→1136.03±86.78 | →1118.46±23.25 | 979.3±33.4→1148.12±14.65 |

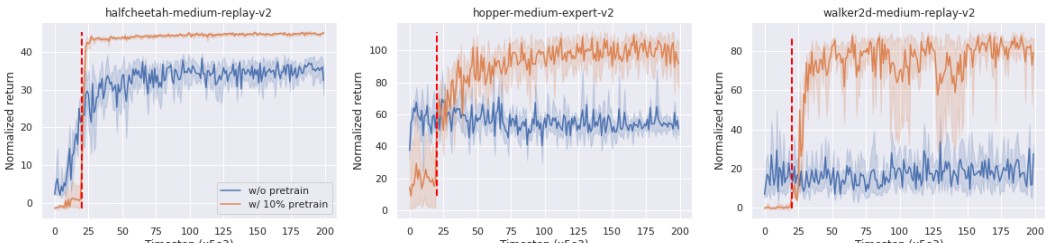

Figure 3: **Learning curves of TD3+BC.** The blue and orange curves are, respectively, the normalized scores of TD3+BC and TD3+BC pretrained with the proposed method. The vertical red reference lines split the pretraining and main training phases. After the pretraining phase, TD3+BC combined with the proposed method quickly outperforms the vanilla TD3+BC by a large margin.

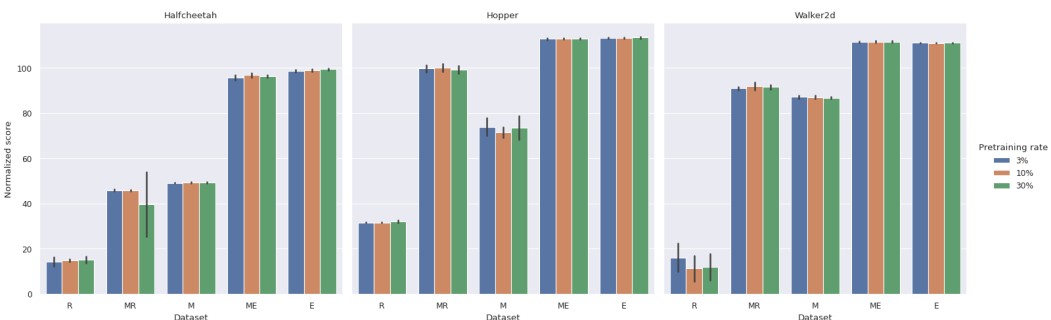

Figure 4: **Averaged normalized scores across pretraining time-step rates.** R, M, MR, ME, and E represent random, medium, medium replay, medium expert, and expert datasets on the D4RL benchmark, respectively.

## 5.1 PERFORMANCE IMPROVEMENT IN OFFLINE RL BENCHMARKS

To demonstrate the effectiveness of the proposed method over existing offline RL methods, we evaluate our method on D4RL and Robomimic datasets. In Table 2, the normalized scores between the vanilla and the one combined with our method are compared for each environment and dataset in D4RL. One can observe that the proposed method combined with the baselines improves the corresponding original methods, achieving an average improvement of $\mathbf{135.94\%}$, across diverse environments and datasets. Specifically, one can observe that all methods including AWAC ($+306.45\%$), CQL ($+132.77\%$), IQL ($+9.21\%$), and TD3+BC ($+95.34\%$) exhibit significantly increased performance on average compared to the results reported in the original papers. We have taken all normalized scores of TD3+BC, AWAC, CQL, IQL from the reported scores in each paper (Fujimoto & Gu, 2021; Nair et al., 2020; Kumar et al., 2020b; Kostrikov et al., 2021a).

Figure 3 shows the learning curves of TD3+BC and the results verify the effectiveness of the proposed method. After the pretraining period (indicated by the red vertical lines), one can notice that the learning curves rapidly increase and achieve higher returns compared to the original methods. These results suggest that our method accelerates training and enhances performance with only a few lines of modifications on top of the baselines. Full graphs of TD3+BC are provided on Figure 12 in Appendix G.

We also applied our method with different pretraining time-step ratios (e.g., 10% - 0.1M of 1M steps) on TD3+BC over 5 seeds. The results are presented on Figure 4. Notably, regardless of the pretraining time-step ratio, the proposed method demonstrates improved performance over different pretraining rates. Overall, the pretraining time-step ratio of 3% yields a slightly higher total sum of averaged scores while the results of the 10% ratio yield the lowest standard deviation. For all of the other experiments in this paper, we use the pretraining time-step ratio of 10%.

Additional experiments are conducted on large-scale robotic manipulation tasks in Robomimic benchmark, to verify the effectiveness of the proposed method for complex tasks. The proposed

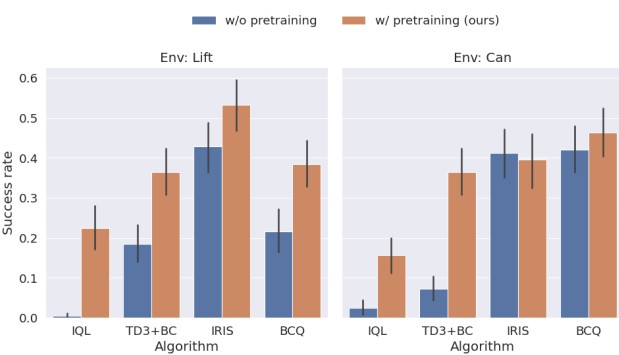

Figure 5: **Averaged success rate on the Robomimic benchmark.** We evaluate both vanilla methods without pretraining (blue) and methods with pretraining (orange). 7 out of 8 cases depict notably improved performance in both environments.

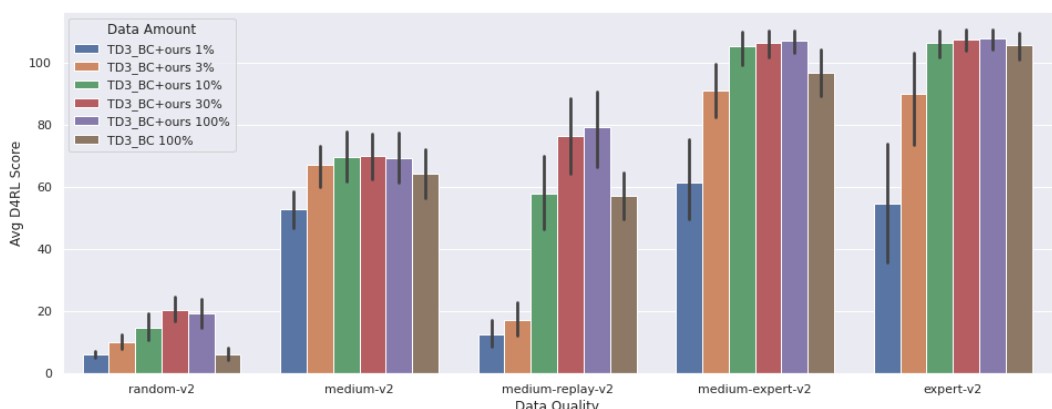

Figure 6: **Averaged normalized scores in reduced datasets across data quality.** This figure shows the overall performance of our method across reduced dataset *(i.e., 1%, 3%, 10%, 30%, 100%)* for three environments *(i.e., halfcheetah, hopper, walker2d)* in D4RL. From the overall results, we conclude that our method guarantees better performance even in 10% of the datasets regardless of the data quality of the dataset, and even 1% for the *random* datasets and 3% for the *medium* datasets.

method is evaluated with tasks containing suboptimal transitions, where the proposed method improves the baselines on the D4RL benchmark. The averaged success rate of four offline RL baselines is reported in Figure 5 with and without applying the proposed method. As can be seen, all the methods with the proposed pretraining method are improved over the baselines in seven out of eight cases. Therefore, we conclude that the proposed method also effectively performs in solving more complex tasks. We also have conducted experiments on Adroit, 24-DOF environment, in Appendix D. The results also demonstrate that the proposed method is effective in solving complex tasks.

### 5.2 DATA EFFICIENCY ACROSS THE QUALITIES OF THE DATASETS

To validate the data efficiency of the proposed method, regardless of the dataset quality, we have examined the proposed method with TD3+BC in reduced datasets *(i.e., 1%, 3%, 10%, 30%, 100% of each dataset)* across the data quality *(i.e., random, medium, medium replay, medium expert, expert)* on D4RL over 5 seeds. To construct the reduced datasets, we have uniformly sampled the transition segments *(i.e., $(s, a, r, s')$)* from each dataset. On the *random* datasets (a leftmost section in Figure 13), training with the proposed method with only 1% of the dataset outperforms the vanilla TD3+BC trained with full datasets at *halfcheetah* and *warker2d* environments. On the *medium* datasets (right to the *random* in Figure 13), the proposed method shows similar or improved results

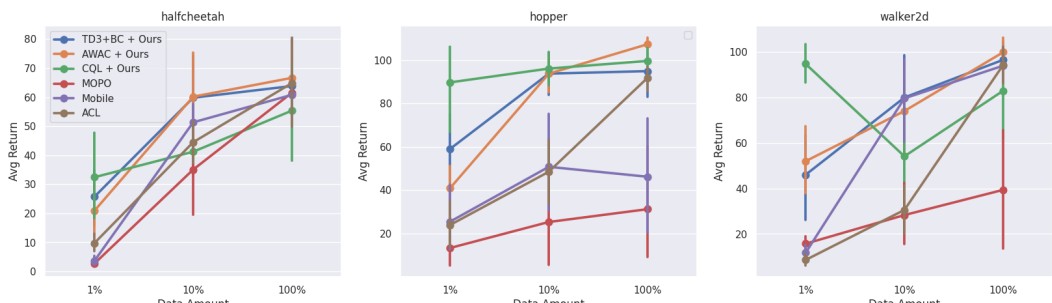

Figure 7: **Comparison of the proposed method with the other approaches on D4RL.** We compare existing model-free offline RLs with our method to offline model-based RLs (*i.e., MOPO, MOBILE*) and a representation RL (*i.e., ACL*) on D4RL over 3 seeds. The graph shows integrated results over *medium, medium-replay, medium-expert* datasets. The results show that the proposed method maintains the performance in reduced datasets, especially 1%, unlike the other approaches.

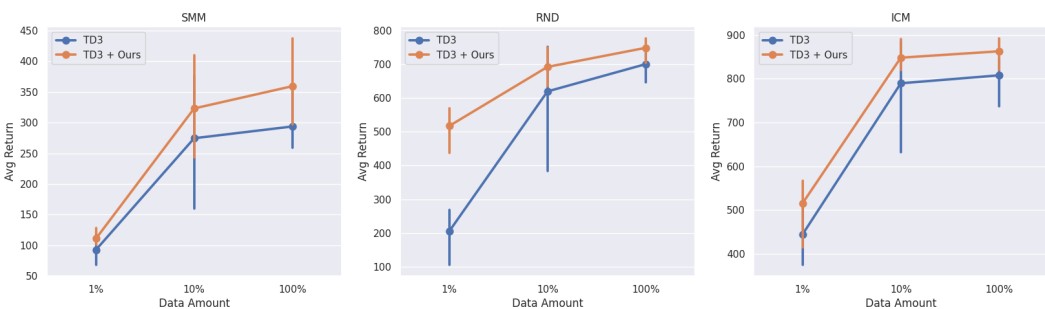

Figure 8: **Average returns in reduced datasets across the dataset collection strategies.** We evaluate our method over different dataset collection strategies *(i.e., SMM, RND, ICM)*. TD3 with our method outperforms the vanilla TD3 overall and even training with 10% of datasets outperforms the vanilla TD3 with full datasets. From the results, we demonstrate that our method is data-efficient regardless of the data distributions.

compared to the vanilla TD3+BC with full datasets by only using 3% of the datasets. On the other datasets (i.e. *medium-replay*, *medium-expert*, and *expert*), the proposed method with 10% datasets totally outperforms the vanilla TD3+BC with full datasets. From the overall results in Figure 6, we conclude that our method guarantees better performance even in 10% of the datasets regardless of the data quality of the dataset.

We also compare our method with offline model-based RL and representation approaches. We apply our method to TD3+BC, AWAC, and CQL. We adopt MOPO (Yu et al., 2020) and MOBILE (Sun et al., 2023) as representatives of offline model-based RL, ACL (Yang & Nachum, 2021) as a representation representative. We conduct the experiments on D4RL, *medium, medium-replay, medium-expert* datasets over three seeds. Figure 7 shows integrated results over the datasets and Figure 14 shows details. The results show that our method maintains the performance in reduced datasets compared with the other approaches that spend extra training budget (e.g., training and forwarding the transition). Especially in 1% datasets, CQL with our method largely outperforms the others. As a result, we claim that our method is the most proper choice for data-efficient offline RL.

## 5.3 DATA EFFICIENCY ACROSS THE DATA DISTRIBUTIONS

We assume that a small dataset would have a shifted distribution compared to a large dataset, for instance, some small datasets have narrow support of visited states. Based on the assumption we have made, we evaluate our method across different dataset collection strategies since each dataset has a different data distribution. In ExoRL (Yarats et al., 2022), we chose TD3 as a comparison algorithm and SMM (Lee et al., 2019), RND (Burda et al., 2018), and ICM (Pathak et al., 2017), as *walker*

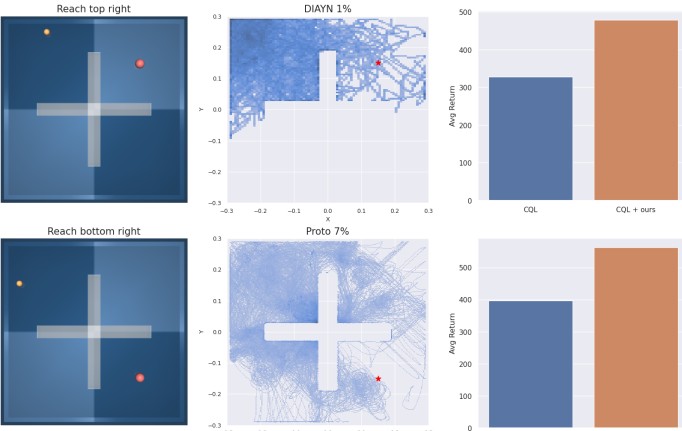

Figure 9: **Effectiveness of the proposed method over narrow support of the visited states datasets.** (Left) Visualized goal-reaching point mass agents and trajectories with different goals, portions, and exploration methods. (Right) Averaged return of CQL trained with two datasets with and without the proposed pretraining method.

*walk* task datasets. In Yarats et al. (2022), ICM shows the best performance, followed by RND, SMM, and TD3 shows the best performance in ICM. We compare TD3 to TD3 with our method in reduced datasets (*i.e., 1%, 10%, 100%)* over three seeds. To construct reduced datasets, we select the data from the front. Figure 8 shows the results. For all datasets, applying our method with only 10% of datasets outperforms vanilla TD3 with full datasets. Especially in RND, even training with 1% of datasets shows a significantly high average return.

Furthermore, we consider a *point mass maze* environment in ExoRL to investigate whether our method is effective even in narrow support of the visited states datasets. Figure 9 visualizes the trajectories of each reduced dataset collected by DIAYN (Eysenbach et al., 2018), and Proto (Yarats et al., 2021a) strategies *(i.e., 1% of DIAYN, 7% of Proto)*. In comparison with Figure 2 in Yarats et al. (2022), our reduced dataset settings cover more narrow support of visited states. The top right figure of DIAYN shows that there are a few trajectories around the *top right goal* and the bottom left right figure of Proto also shows that there are a few trajectories around the *bottom right goal* in Figure 9. To demonstrate our method is effective even with a dataset with this shifted state distribution, we evaluated the proposed method on reduced *point mass maze* datasets described in Figure 9 over short (*reach top right*) and long (*reach bottom right*) goals with CQL. Figure 9 demonstrates that our method shows significant performance even with narrow data distribution. From the results, we conclude that our method is indeed more data-efficient than the other methods regardless of different choices of the data distribution.

## 6 CONCLUSION

In this paper, we propose a simple yet effective data-efficient offline RL method that pretrains a shared $Q$-network with the transition dynamics prediction task, maintaining reasonable performance even with a small training dataset. To pretrain the $Q$-network, we design a novel shared network architecture that outputs predictions of the next state and Q-value. This structure makes our method easy to apply to any existing offline RL algorithms and efficiently boosts data efficiency.

To demonstrate the effectiveness of the proposed strategy, we conduct experiments with various settings in offline RL. From the results, we demonstrate that our method significantly improves the performance of existing offline RL algorithms over D4RL and Robomimic benchmarks. We also demonstrate that our method is indeed data-efficient across the different data qualities from D4RL and the different data distributions from ExoRL. We leave future work to expand our method toward various offline RL problems, e.g., offline to online RL, goal-conditioned RL, and real-world applications.

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

# A  PRETRAINING Q-NETWORK FOR ONLINE RL (OFF-POLICY)

---

**Algorithm 2** Pretraining phase for Online RL (Off-policy)

---

**Input**: Learning rate $\alpha$
Initialize parameters $\varphi, \psi$ and a buffer $\mathcal{D}$
**for** each gradient step **do**
  Uniformly sample a random action and collect a transition
  $a \sim U(a_{min}, a_{max})$
  $s' \sim p(s'|s, a)$
  Update the buffer with a collected transition
  $\mathcal{D} \leftarrow \mathcal{D} \cup \{(s, a, r, s')\}$

  Sample a mini-batch $\mathcal{B} \sim \mathcal{D}$
  Compute the forward dynamics prediction error

$$\mathcal{L}_{pre}(\varphi, \psi) = \sum_{(s,a,s') \in \mathcal{B}} (s' - (g_\psi \circ h_\varphi)(s, a))^2$$

  Update weights of the shared network and forward network

$$\varphi \leftarrow \varphi - \alpha \nabla_\phi \mathcal{L}_{pre}(\varphi, \psi), \quad \psi \leftarrow \psi - \alpha \nabla_\psi \mathcal{L}_{pre}(\varphi, \psi)$$

**end for**
**Output**: Pretrained weights $\varphi$ of the shared network, collected buffer $\mathcal{D}$

---

---

**Algorithm 3** Pretraining phase for Online RL (Off-policy) with pre-collected dataset

---

**Input**: Dataset $\mathcal{D}_{pre}$ of transition $(s, a, s')$, Learning rate $\alpha$
Initialize parameters $\varphi, \psi$
**for** each gradient step **do**
  Sample a mini-batch $\mathcal{B} \sim \mathcal{D}_{pre}$
  Define the loss function

$$\mathcal{L}_{pre}(\varphi, \psi) = \sum_{(s,a,s') \in \mathcal{B}} (s' - (g_\psi \circ h_\varphi)(s, a))^2$$

  Take the gradient descent step

$$\varphi \leftarrow \varphi - \alpha \nabla_\phi \mathcal{L}_{pre}(\varphi, \psi), \quad \psi \leftarrow \psi - \alpha \nabla_\psi \mathcal{L}_{pre}(\varphi, \psi)$$

**end for**
**Output**: Pretrained weights $\varphi$ of the shared network

---

We extended our pretraining method to popular online off-policy RL methods by incorporating the pretraining phase ahead of the main training phase. During the pretraining phase of the online agent, a trajectory dataset was obtained by either initializing the replay buffer with actively collected interaction data by uniformly sampling a random action or offline static dataset.

For experiments on online RL using an off-policy setting, we adopted soft actor-critic (SAC) Haarnoja et al. (2018) and twin delayed deep deterministic policy gradient algorithm (TD3) Fujimoto et al. (2018). We compare these algorithms with and without our pretraining method on OpenAI Gym MuJoCo tasks. For a fair comparison, all algorithms were trained for 1 million time steps on each task over 5 seeds.

Table 3 presents the results of the experiments following Algorithm 2 which collects the pretraining dataset by uniformly sampling random actions. Incorporating our pretraining phase shows better performance in more than half of the results. Additionally, we trained both SAC and TD3 with the pre-collected dataset from the D4RL for the pretraining phase along the Algorithm 3. Note that we only used the pre-collected dataset during the pretraining phase. Table 4 shows the best scores among the 5 datasets (i.e., random, medium, medium replay, medium expert, expert). Interestingly,

pretraining with the suboptimal-level dataset (medium-replay) shows better performance compared to the expert-level dataset.

Table 3: Results of Off-policy RL application on OpenAI gym MuJoCo tasks

|  | SAC | TD3 |
|---|---|---|
| HalfCheetah-v2 | 10065.77±621.80→11005.51±374.14 | 10644.63±190.42→11697.71±236.01 |
| Hopper-v2 | 3357.07±30.64→1419.55±137.55 | 3365.08±94.69→3454.83±129.34 |
| Walker2d-v2 | 4279.67±509.51→2697.92±674.29 | 4193.11±435.31→4481.19±190.93 |
| Ant-v2 | 4191.17±986.11→4399.56 766.24 | 5172.78±659.02→4407.40±759.64 |
| Humanoid-v2 | 5545.70±85.00→479.09 83.86 | 5247.14±187.64→5816.16±199.25 |
| Pusher-v2 | -190.77±88.51→-133.96 29.00 | -22.94±0.52→-22.85±1.25 |

Table 4: Results of Off-policy RL pretrain with the D4RL OpenAI gym MuJoCo datasets

|  | SAC | TD3 |
|---|---|---|
| HalfCheetah-v2 | 10402.79±1675.67 | 11820.06±269.76 |
| Hopper-v2 | 3405.95±70.87 | 3465.25±149.87 |
| Walker2d-v2 | 4785.15±247.37 | 4559.38±1007.69 |

From the above experiments, we conjecture that pretrained online RL (off-policy) has limitations when they only exploit random action data for pretraining. A marginal state distribution induced by uniformly sampling random actions is close to the initial state distribution, limiting the diversity in the dataset and eventually leading to an increase in forward dynamics uncertainty. Consequently, there are fewer opportunities to learn the good features of forward dynamics with random action datasets than suboptimal-level datasets. This explains why Table 3 shows worse results than Table 4.

We also applied another approach introduced in section B to online RL settings. The results, shown in Table 5, indicate that more than half exhibit enhanced performance compared to reported scores in Table 3.

Table 5: Results of Off-policy RL with Additional Loss

|  | SAC | TD3 |
|---|---|---|
| HalfCheetah-v2 | 8498.68±3195.13 | 9588.53±866.30 |
| Hopper-v2 | 3539.39±133.47 | 3523.67±202.52 |
| Walker2d-v2 | 4847.86±135.52 | 3819.68±552.84 |
| Ant-v2 | 3710.73±917.35 | 5401.0±844.56 |
| Humanoid-v2 | 5576.98±106.31 | 5489.73±38.28 |
| Pusher-v2 | -158.66±55.02 | -25.47±34.00 |

## B ANOTHER DESIGN CHOICE USING OUR SHARED Q-NETWORK STRUCTURE

In this section, we introduce another approach that also utilizes features of forward dynamics using the shared networks as in the previous pretraining method. In this approach, we use the following modified loss that adds the forward model loss to the loss for the $Q$-function estimation:

$$\mathcal{L}_Q = \mathcal{L}_{TD} + \mathcal{L}_{dynamics} \tag{6}$$

In this way, the shared network is trained throughout the entire training period without the pretraining phase. We adopt TD3+BC for evaluation and the results are presented in table 6. On TD3+BC, this approach also outperforms almost all of the vanilla scores. Simply adding the supervised loss term of state prediction without any multiplier or technique demonstrates improved performance. Consequently, we suggest that the proposed shared Q-network can be expanded in other directions and we expect that it holds significant potential for further research.

Table 6: **Averaged normalized scores of TD3+BC with additional loss on D4RL benchmark.** We depict increased scores compared to their original scores in blue color and report mean and standard deviations over 5 random seeds.

|  | Random | Medium | Medium Replay | Medium Expert | Expert |
|---|---|---|---|---|---|
| HalfCheetah-v2 | 11.45±0.51 | 48.23±0.33 | 44.93±0.29 | 93.55±1.00 | 96.59±0.25 |
| Hopper-v2 | 31.54±0.42 | 70.86±2.17 | 90.39±7.34 | 113.44±0.35 | 113.28±0.20 |
| Walker2d-v2 | 13.46±6.58 | 82.65±1.65 | 86.11±1.54 | 111.88±0.63 | 110.98±0.22 |

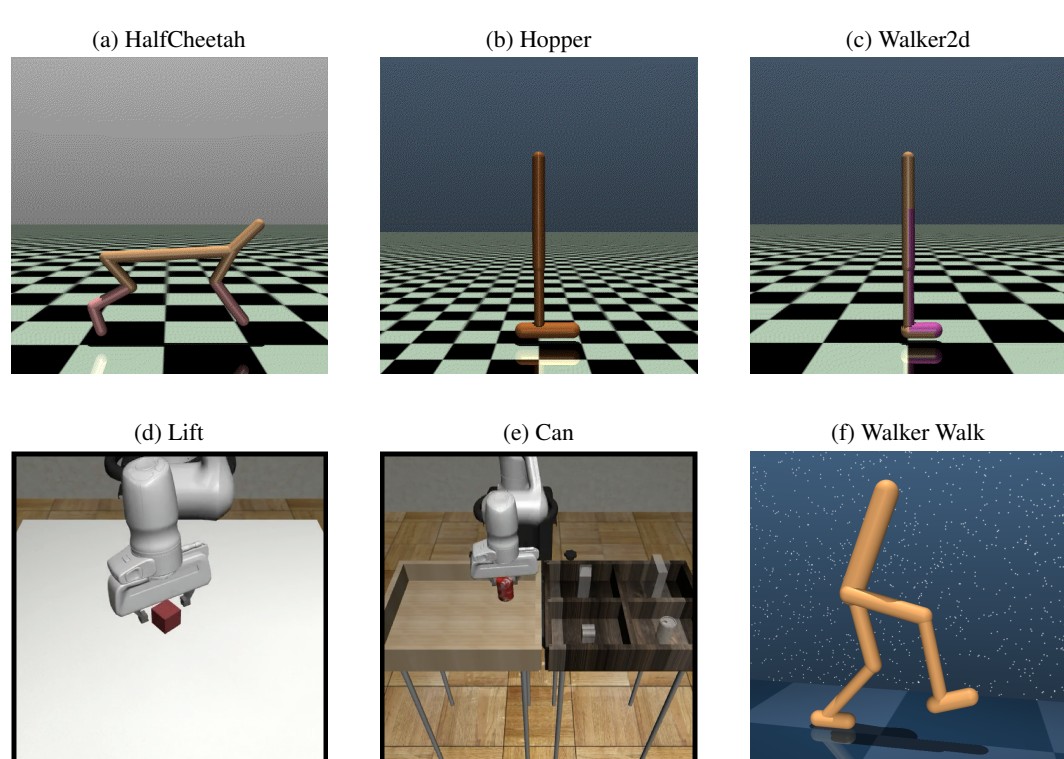

Figure 10: **Illustrations of environments.**

## C  TASKS AND DATASETS

In this section, we provide detailed experimental setups for the tasks and datasets. Illustrated environments can be found in Figure 10

### C.1  D4RL

D4RL consists of 8 separate tasks. In this work, we utilized one of them for the main experiments; OpenAI Gym MuJoCo continuous control tasks. It consists of 4 different environments (i.e., HalfCheetah, Walker2d, Hopper, and Ant) and 5 heterogeneous datasets in terms of data quality for each environment. Each dataset is collected along the below strategies:

- Random (1M samples): Collected from a randomly initialized policy.
- Expert (1M samples): Collected from a policy trained to completion with SAC.
- Medium (1M samples): Collected from a policy trained to approximately 1/3 the performance of the expert.
- Medium-Expert (almost 2M samples): A 50-50 split of medium and expert data.

- Medium-Replay (almost 3M samples): Collected from the replay buffer of a policy trained up to the performance of the medium agent.

All environments have the same episode limit of 1000 and the goal of each locomotion agent is to run as fast as possible without falling to the ground. More detailed information can be found at https://github.com/Farama-Foundation/D4RL.

### C.2 ROBOMIMIC

Robomimic provides a large-scale and diverse collection of task demonstrations spanning multiple human or robotic demonstrations of varying quality. We considered machine-generated (MG) datasets generated by training an SAC agent for each task and then using intermediate policies to generate mixed-quality datasets. We selected this dataset for evaluation since our method demonstrated superior performance with suboptimal datasets on the D4RL benchmark. All environments have the same episode limit of 400. The goal of the Lift environment is lifting the cube above a certain height and the goal of the Can environment is placing the can into the corresponding container. More detailed information can be found at https://github.com/ARISE-Initiative/robomimic.

### C.3 EXORL

They provide exploratory datasets for 6 DeepMind Control Stuite domains (*i.e., Cartpole, Cheetah, Jaco Arm, Point Mass Maze, Quadruped, Walker*) and totally 19 tasks. For each domain, they collected datasets by running 9 unsupervised RL algorithms (*i.e., APS, APT, DIAYN, Disagreement, ICM, ProtoRL, Random, RND, SMM*) from URLB for total of 10M steps. More detailed information can be found at https://github.com/denisyarats/exorl?tab=readme-ov-file.

## D EXPERIMENTS ON ADROIT IN D4RL

We conducted additional experiments on adroit in D4RL Fu et al. (2020) benchmark to validate that our method can be adopted to different complex domains. An illustration of the Adroit environment can be found in Figure 11. The Adroit domain involves controlling a 24-DoF robotic hand with 4 different control tasks (i.e., Pen, Door, Hammer, and Relocate) and 3 heterogeneous datasets as following:

- Human: Collected with the 25 human demonstrations provided in the DAPG Rajeswaran et al. (2017) repository.
- Cloned: a 50-50 split between demonstration data and 2500 trajectories sampled from a behavioral cloned policy on the demonstrations. The demonstration trajectories are copied to match the number of behavioral cloned trajectories.
- Expert: Collected with 5000 trajectories sampled from an expert that solves the task, provided in the DAPG repository.

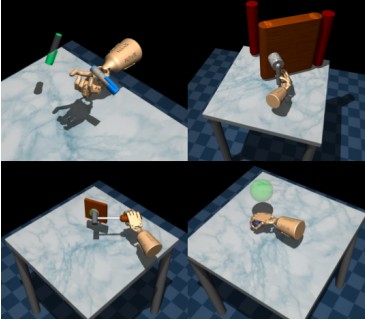

Figure 11: **The tasks of Adroit. (top left)** Pen - aligning a pen with a target orientation, **(top right)** Door - opening a door, **(bottom left)** Hammer - hammering a nail into a board, **(bottom right)** Relocate - moving a ball to a target position.

For experiments, we compared AWAC, IQL, and TD3+BC with/without our pretraining method over 5 seeds. Table 7 yields averaged normalized scores for each task. Overall, learning with our pretraining phase demonstrates enhanced performance. From these results, we conclude that our method can be effective in complex domains not only tabletop but dexterous manipulation as well.

Table 7: **Averaged normalized scores on Adroit.** Left-hand side scores are scores of vanilla methods. Right-hand side scores are scores of baselines combined with our pretraining method. We depict increased scores compared to their original scores in blue color and report mean and standard deviations over 5 random seeds.

|  |  | AWAC | IQL | TD3+BC |
|---|---|---|---|---|
| Human | Pen | 146.19±5.29→157.60±5.28 | 101.87±14.34→104.66±17.30 | 20.32±5.97→20.78±10.93 |
|  | Hammer | 7.98±9.41→36.95±35.13 | 14.33±5.22→17.78±9.27 | 2.40±0.16→2.38±0.17 |
|  | Door | 60.82±12.38→29.96±22.43 | 6.74±1.31→5.81±3.20 | -0.09±0.00→-0.04±0.04 |
|  | Relocate | 1.51±1.05→3.91±2.21 | 1.20±1.05→1.52±1.11 | -0.29±0.01→-0.18±0.13 |
| Cloned | Pen | 145.37±4.19→144.48±3.42 | 98.38±16.13→97.76±16.90 | 39.69±18.95→48.18±11.27 |
|  | Hammer | 10.37±7.88→12.61±8.66 | 8.94±2.07→11.38±4.46 | 0.59±0.17→1.17±0.61 |
|  | Door | 2.95±2.97→9.59±7.73 | 5.61±3.02→5.00±1.44 | -0.23±0.11→-0.03±0.03 |
|  | Relocate | 0.04±0.09→0.18±0.21 | 0.91±0.45→1.06±0.40 | -0.02±0.09→-0.13±0.09 |
| Expert | Pen | 163.99±1.19→163.73±1.88 | 148.38±2.46→147.79±3.06 | 131.73±19.15→141.10±10.28 |
|  | Hammer | 130.08±1.30→130.04±0.48 | 129.46±0.42→129.50±0.36 | 33.36±34.61→59.76±52.35 |
|  | Door | 106.67±0.28→106.95±0.16 | 106.45±0.29→106.71±0.28 | 0.99±0.83→0.87±1.48 |
|  | Relocate | 109.70±1.32→111.27±0.35 | 110.13±1.52→109.82±1.45 | 0.57±0.33→0.22±0.13 |
| Total |  | 885.67±47.35→907.26±87.94 | 732.40±48.27→738.79±59.23 | 229.03±80.40→274.08±87.49 |

## E  IMPLEMENTATION DETAILS

In this section, we provide detailed implementation setups for extensive experiments. Since we suggest a plug-and-play pretraining method for popular offline RL methods, we reuse open-source code for comparative results: TD3+BC[1], IQL[2], AWAC[3], and CQL[4] for D4RL. We use off-the-shelf offline methods in the official repository[5] for the Robomimic environment. We only use open-source baselines which use PyTorch for fair comparisons. On the D4RL, we train each agent with 1M gradient steps for each environment over 5 seeds. Also, we evaluate each agent with 5 rollouts every 5k gradient steps for TD3+BC, AWAC, and CQL and 10k gradient steps for IQL. We report the best scores for all tables and figures. On the Robomimic, we train each agent with 200k gradient steps for each environment over 5 seeds. Also, we evaluate each agent with 50 rollouts over 5 seeds. For all experiments, we used RTX-A5000 GPU for training and evaluation.

## F  DISCUSSIONS

In this section, we address the potential concerns regarding our method's novelty since it closely connects with prior approaches in relevant fields. We provide our detailed discussions in separate subsections of each topic.

**Representation Learning.** Over recent years, the field has observed a significant amount of literature working on predictive representation in RL. Concerning the similarity with prior works, we claim that the idea of pretraining shared Q-network for improving data efficiency is remarkable. Our method pretrains the neural networks with the next state prediction objective to improve an underlying RL agent's performance and data efficiency similar to (Schwarzer et al., 2020; Guo et al., 2018). However, Schwarzer et al. (2020) has proposed an online training method in a self-supervised learning manner whereas our method considers supervised learning for pretraining. Since the self-predictive task in Schwarzer et al. (2020) is conducted in latent space, representation learning is essentially involved with the task.

---

[1] https://github.com/sfujim/TD3_BC
[2] https://github.com/Manchery/iql-pytorch
[3] https://github.com/hari-sikchi/AWAC
[4] https://github.com/young-geng/CQL
[5] https://github.com/ARISE-Initiative/robomimic

Therefore, adopting advanced training techniques including data augmentation (Yarats et al., 2021b) and the use of a target encoder (He et al., 2020) significantly affect the RL agent's performance. Additionally, Schwarzer et al. (2020) suggests a self-supervised representation learning with the latent transition prediction task in the online RL regime. In comparison, our method alleviates an introduction of extra techniques other than the shared network architecture, proving superior performance in offline RL benchmarks of diverse environments, e.g. locomotion and manipulation tasks.

Guo et al. (2018) has presented an unsupervised learning method that encodes the *belief state* capturing sufficient information of the hidden true state from a past interaction history. In other words, the main interest of Guo et al. (2018) is how the neural network architecture trained with unsupervised learning extracts adequate information concerning the true state in POMDP, not how the underlying RL method given rich representation performs decision-making problem well. Specifically, the network architecture in Guo et al. (2018) is based on GRU, RNN based sequential network, and predicts a next observation $o_{t+1}$ using action $a_t$ and a belief state $b_t$ that contains the partial information of the previous trajectory. Conversely, our method is implemented on MLP with the shared network architecture and predicts the next state $s_{t+1}$ using current state $s_t$ and action $a_t$ without a past history.

**Model-based RL.** One might argue that our method lacks novelty with the idea of training a neural network with the transition dynamics prediction task. Obviously, the idea of approximating the transition dynamics (Sutton, 1991) for downstream RL training is not what we first suggest. However, we contend that our method has a few refuting viewpoints with previous similar works. TDMPC (Hansen et al., 2022) and TDMPC2 (Hansen et al., 2023) are model-based single and multi-task RL approaches, which recursively feed the output of the same network (i.e. the encoder and task embedding network) for the transition model and value learning. The outputs of the shared backbone networks correspond to the latent representation and task embedding vector, respectively, and most latent model-based RL approaches including TDMPC reuse the outputs for the transition model and value learning. On the other hand, our method presents a shared network architecture resembling the dueling architecture (Wang et al., 2016) to pretrain the shared backbone network with a separated stream (a header) of the transition model and Q-network. Additionally, this paper presents a two-phase training scheme: the transition model combined with the shared network is trained with the transition dynamics prediction task in the first phase and the Q-network, consisting of an MLP header and the shared network initialized with the parameter of the shared network in the first phase, is trained with the downstream RL value learning task in the second phase.

JOWA (Cheng et al., 2024) is an offline world model for multi-task RL with a shared Transformer backbone network for sequential a next-token prediction task. By modeling the decision-making problem to the sequential token prediction task, the backbone network, tokenizer, and header are trained in a supervised manner with the offline dataset. While the main purpose of JOWA is scaling an offline world model across multiple tasks with generalized performance over unseen tasks, this paper intends to improve the data efficiency of conventional offline RL approaches in single-task RL. Furthermore, our method alleviates additional training after offline RL training with a novel two-phase training strategy while JOWA allows few-shot fine-tuning for sample efficient transfer with a multi-game environment. Even with a similar purpose of data efficiency, our method entails a minimal algorithmic change with a consistent training budget compared to previous approaches.

Dreamer (Hafner et al., 2023) has brought a notable advancement in model-based RL. Dreamer suggests a world model for decision-making with a considerate design of the latent transition model and reconstructive objective. Since jointly learning an accurate world model and actor in a multi-task environment is challenging, the expensive cost of collecting samples often becomes problematic. In contrast, our method does not necessitate extra modifications of conventional offline RL and proves its sufficient performance gains in comprehensive experiments. Considering previous improvements in representation learning usually involve state-of-the-art design choices (e.g. data augmentation), this paper would contribute to reasonable architectural achievements for researchers by presenting a minimal training structure with verified performance profit.

# G  LEARNING CURVES

In this section, we provide the full results of learning curves in the section 5.1 for further information.

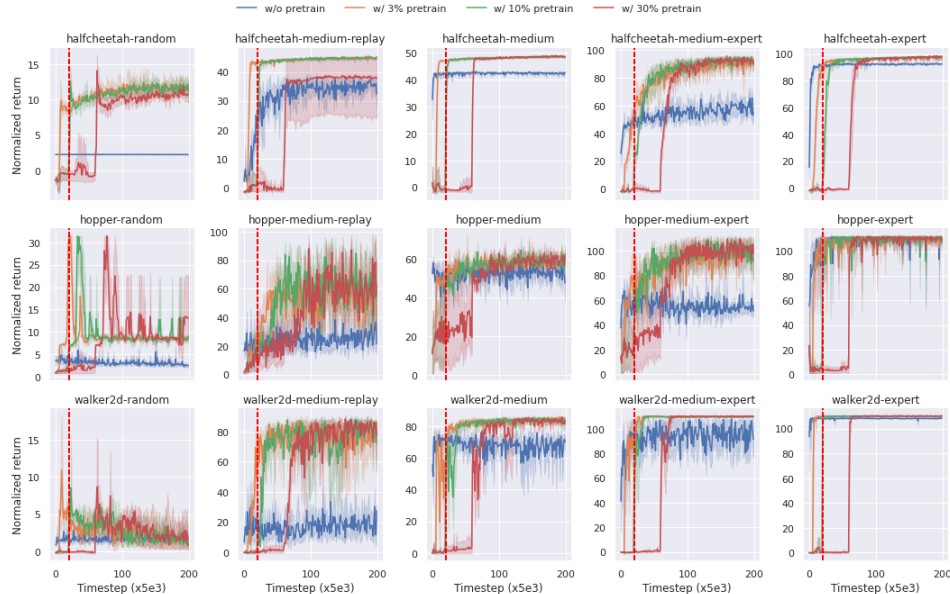

Figure 12: **Learning curves of TD3+BC on the D4RL benchmark.**

# H  EXPERIMENTS WITH LINEAR APPROXIMATED $Q$-NETWORK

In this section, We pretrained TD3+BC and froze it except for the last linear layer during the remaining learning time. The blue-colored scores indicate improved scores from the reported scores from the original TD3+BC. Although only the last linear layer of the pretrained TD3+BC was trained and the shared network was frozen, it shows better performance than the vanilla CQL. Moreover, it shows better performance than the others over the suboptimal level of the datasets (i.e., random, medium, medium replay).

Table 8: **Results of pretrained TD3+BC which approximated with linear $Q$ function.**

|  |  | AWAC | CQL | IQL | TD3+BC | freezed TD3+BC |
|---|---|---|---|---|---|---|
| Random | HalfCheetah | 2.2 | 21.7±0.9 |  | 10.2±1.3 | 6.03±2.65 |
|  | Hopper | 9.6 | 10.7±0.1 |  | 11.0±0.1 | 11.59±10.56 |
|  | Walker2d | 5.1 | 2.7±1.2 |  | 1.4±1.6 | 7.18±0.58 |
| Medium | HalfCheetah | 37.4 | 37.2±0.3 | 47.4 | 42.8±0.3 | 42.64±1.19 |
|  | Hopper | 72.0 | 44.2±10.8 | 66.4 | 99.5±1.0 | 67.16±3.56 |
|  | Walker2d | 30.1 | 57.5±8.3 | 78.3 | 79.7±1.8 | 72.03±0.78 |
| Medium Replay | HalfCheetah |  | 41.9±1.1 | 44.2 | 43.3±0.5 | 40.21±0.79 |
|  | Hopper |  | 28.6±0.9 | 94.7 | 31.4±3.0 | 64.41±19.54 |
|  | Walker2d |  | 15.8±2.6 | 73.9 | 25.2±5.1 | 41.02±12.05 |
| Medium Expert | HalfCheetah | 36.8 | 27.1±3.9 | 86.7 | 97.9±4.4 | 47.35±8.73 |
|  | Hopper | 80.9 | 111.4±1.2 | 91.5 | 112.2±0.2 | 95.07±15.27 |
|  | Walker2d | 42.7 | 68.1±13.1 | 109.6 | 101.1±9.3 | 74.75±0.59 |
| Expert | HalfCheetah | 78.5 | 82.4±7.4 |  | 105.7±1.9 | 61.93±10.71 |
|  | Hopper | 85.2 | 111.2±2.1 |  | 112.2±0.2 | 113.13±0.39 |
|  | Walker2d | 57.0 | 103.8±7.6 |  | 105.7±2.7 | 57.14±44.96 |
| Total |  |  | 764.3±61.5 |  | 979.3±33.4 | 801.64±132.34 |

# I EXPERIMENTS WITH VARIOUS AMOUNT OF DATA

In this section, we provide more details in section 5.2 of the dataset size. We conducted each experiment with the same settings in subsection 5.1 over 5 seeds and reported the results that exhibit averaged normalized scores.

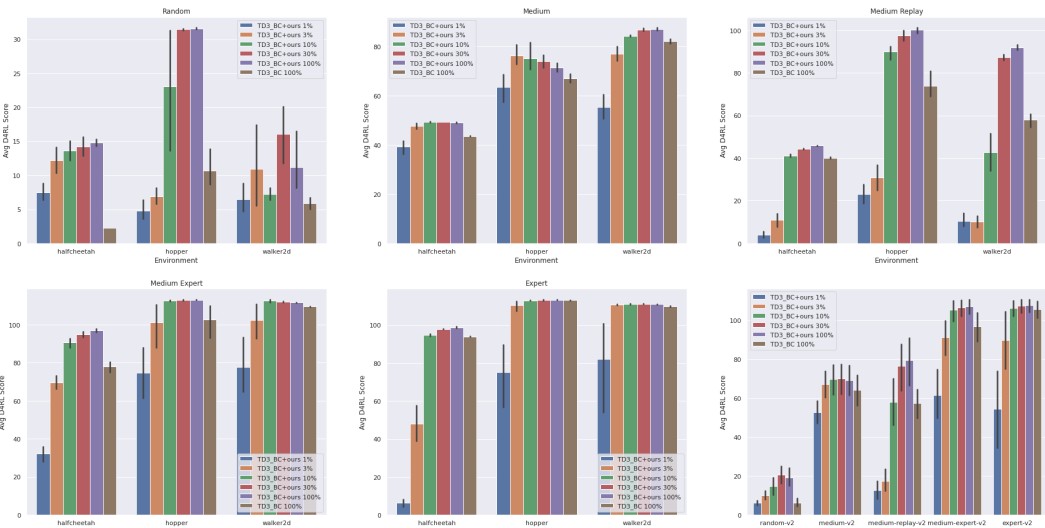

Figure 13: **Averaged normalized scores across dataset optimal quality and sizes.** This figure compares the performance of our method with TD3+BC in reduced datasets *(i.e., 1%, 3%, 10%, 30%, 100% of each dataset)* to vanilla TD3+BC across the data quality *(i.e., random, medium, medium replay, medium expert, expert)* on D4RL. From the overall results (Bottom Right), we conclude that our method guarantees better performance even in 10% of the datasets regardless of the data quality of the dataset.

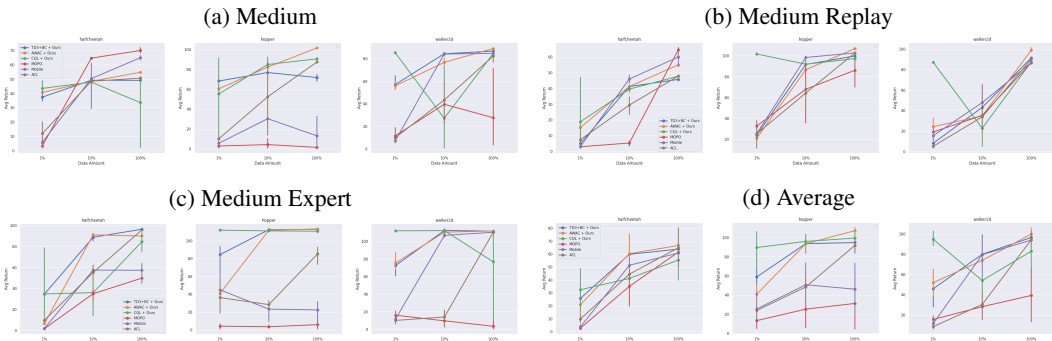

Figure 14: **Comparison with offline model-based RL and representation approaches.** We compare TD3+BC, AWAC, CQL with ours to offline model-based RLs *(i.e., MOPO, Mobile)* and a representation RL *(i.e., ACL)* on D4RL over 3 seeds. The gragh shows results over *medium, medium-replay, medium-expert* datasets. The results show that our method maintains the performance in reduced datasets, especially 1%, unlike the other approaches.

Table 9: **Results of pretrained AWAC over various size.**

| | | w/o pretrain | w/ pretrain 10% | w/ pretrain 30% | w/ pretrain |
|---|---|---|---|---|---|
| Random | HalfCheetah | 2.2 | 9.71±3.08 | 36.37±1.47 | 51.10±0.89 |
| | Hopper | 9.6 | 97.05±3.24 | 93.35±6.32 | 59.47±33.79 |
| | Walker2d | 5.1 | 8.57±0.47 | 8.36±1.30 | 13.11±3.91 |
| Medium | HalfCheetah | 37.4 | 55.47±1.52 | 56.64±2.68 | 54.63±1.45 |
| | Hopper | 72.0 | 101.28±0.78 | 101.32±0.20 | 101.73±0.20 |
| | Walker2d | 30.1 | 95.14±1.46 | 91.38±1.37 | 89.51±0.88 |
| Medium Replay | HalfCheetah | | 51.00±0.69 | 52.12±0.76 | 55.75±1.30 |
| | Hopper | | 103.67±1.81 | 107.69±1.71 | 106.67±0.59 |
| | Walker2d | | 104.10±1.57 | 105.42±1.97 | 100.31±2.11 |
| Medium Expert | HalfCheetah | 36.8 | 83.18±1.69 | 86.55±0.94 | 90.05±1.89 |
| | Hopper | 80.9 | 113.01±0.71 | 113.34±0.09 | 113.23±0.22 |
| | Walker2d | 42.7 | 117.26±1.77 | 114.68±2.18 | 111.88±0.28 |
| Expert | HalfCheetah | 78.5 | 91.54±1.04 | 93.46±0.54 | 93.48±0.11 |
| | Hopper | 85.2 | 113.02±0.17 | 113.18±0.20 | 112.86±0.10 |
| | Walker2d | 57.0 | 117.92±2.07 | 112.55±0.56 | 111.22±0.35 |
| Total | | | 1261.90±22.05 | 1286.43±22.28 | 1265.01±48.07 |

Table 10: **Results of pretrained IQL over varying dataset sizes.**

| | | w/o pretrain | w/ pretrain 10% | w/ pretrain 30% | w/ pretrain |
|---|---|---|---|---|---|
| Random | HalfCheetah | | 6.92±0.63 | 12.65±2.53 | 18.28±1.02 |
| | Hopper | | 8.17±0.54 | 9.93±1.19 | 10.67±0.41 |
| | Walker2d | | 8.26±0.64 | 9.08±0.96 | 8.88±0.71 |
| Medium | HalfCheetah | 47.4 | 46.51±0.18 | 47.87±0.21 | 48.85±0.16 |
| | Hopper | 66.4 | 75.72±3.23 | 80.76±3.51 | 78.62±2.21 |
| | Walker2d | 78.3 | 82.62±1.03 | 83.89±1.69 | 83.63±1.14 |
| Medium Replay | HalfCheetah | 44.2 | 33.49±1.26 | 41.16±0.50 | 45.48±0.17 |
| | Hopper | 94.7 | 80.59±8.25 | 91.08±3.67 | 99.43±1.71 |
| | Walker2d | 73.9 | 39.08±10.42 | 75.33±4.17 | 87.95±1.68 |
| Medium Expert | HalfCheetah | 86.7 | 87.44±2.52 | 93.66±0.46 | 95.25±0.14 |
| | Hopper | 91.5 | 93.89±10.67 | 91.05±18.78 | 105.77±11.31 |
| | Walker2d | 109.6 | 111.23±0.83 | 111.65±0.93 | 112.09±0.93 |
| Expert | HalfCheetah | | 77.85±3.82 | 95.88±0.44 | 97.40±0.13 |
| | Hopper | | 109.16±3.25 | 112.85±1.30 | 113.34±0.46 |
| | Walker2d | | 113.76±2.55 | 112.53±1.35 | 112.80±1.08 |
| Total | | | 974.68±49.84 | 1069.36±41.69 | 1118.46±23.25 |

