# OpenReview forum: "Pretraining A Shared Q-Network for Data Efficient Offline Reinforcement Learning"
_ICLR.cc/2025/Conference — Submitted to ICLR 2025_

### Official Review · Reviewer_DMxg · 2024-10-29

**Soundness:** 3
**Presentation:** 3
**Contribution:** 2
**Rating:** 5
**Confidence:** 4

**Summary:**

The authors propose a method to enhance data efficiency in offline RL by incorporating a pre-training phase, during which the encoded features are used to predict the features of the next state. The paper analyzes why the learned features are well-suited for subsequent RL training and demonstrates performance gains when integrating this pre-training phase with state-of-the-art offline RL methods.

**Strengths:**

1) The paper tries to propose a theoretical analysis on the reason why such a predictive pre-training phase helps.
2) A comprehensive experimental analysis is provided

**Weaknesses:**

1) The concept of learning a predictive representation for RL is not new.
For example, previous papers [Schwarzer et al., 2020, 2021, Hafner et al., 2024] have leveraged predictive representation for RL and demonstrated improved sample efficiency with this approach.
The primary difference between this paper and the papers mentioned above is the evaluation setting:
the proposed method is evaluated under an offline RL setting.
Are there any other key differences I may have overlooked?
2) The analysis of the effect of predictive pre-training is somewhat unclear to me.
If I understand correctly, the logic behind the analysis is as follows:
predictive pre-training $\rightarrow$ high rank feature space $\rightarrow$ smaller error between $C(H_\phi)$ and $Q^\pi$ $\rightarrow$ smaller error between $Q^\pi$ and $Q_{\phi, \theta}$.
- For the first implication, empirical results are presented, but a theoretical analysis is missing.
- Could you provide an explanation for the second implication? A clearer explanation or proof for why a higher rank feature space leads to smaller error would be helpful.
- Could you give a more explicit derivation of how Equation 5 supports the final implication?

I believe the analysis would benefit from greater coherence throughout, which would strengthen the paper.

- M. Schwarzer, A. Anand, R. Goel, R. D. Hjelm, A. C. Courville, and P. Bach-
man. Data-efficient reinforcement learning with self-predictive representa-
tions. In International Conference on Learning Representations, 2020.
- M. Schwarzer, N. Rajkumar, M. Noukhovitch, A. Anand, L. Charlin, D. Hjelm,
P. Bachman, and A. Courville. Pretraining representations for data-efficient
reinforcement learning. In Conference on Neural Information Processing Systems, 2021
- D. Hafner, J. Pasukonis, J. Ba, and T. Lillicrap. Mastering diverse domains
through world models, 2024. URL https://arxiv.org/abs/2301.04104.

**Questions:**

1) Is the encoder fixed after the pre-training or not?
Which one is better?
2) Could you explain more about the percentage ratio/percentage rate (line 362)?
Is the training set separated into two parts?
For example, 5% data is used for pre-training the representation and the other data is used for RL training.

---

> ### Author Response · Authors · 2024-11-19
>
> We thank the reviewer for providing thorough and deliberate reviews of our work. We believe that the reviewer's dedicated considerations and concerns indeed improve the quality and completeness of the paper. We have tried to address the informed concerns below and clarified suggested weaknesses and questions.
>
> # Weaknesses
>
> > The primary difference between this paper and the papers mentioned above is the evaluation setting: the proposed method is evaluated under an offline RL setting.
>
> Over recent years, the field has observed a significant amount of literature working on predictive representation in RL. Suggested works including our work keep delving into data-efficient RL for faster and further deployments of deep RL in the real world. Concerning the similarity with prior works, we claim that the idea of pretraining shared Q-network for improving data efficiency is remarkable.
>
> To begin with, the proposed method pretrains the neural networks with the next state prediction objective to improve an underlying RL agent's performance and data efficiency similar to [1, 2]. However, these works have proposed a representation pretraining method in a self-supervised learning manner whereas the proposed method considers supervised learning for pretraining. Since the self-predictive task in these works is conducted in latent space, representation learning (e.g. contrastive learning) is essentially involved with the task. Therefore, adopting advanced training techniques including data augmentation [3] and the use of a target encoder [4] significantly affect the RL agent's performance. Additionally, these works suggest a self-supervised representation learning with the latent transition prediction task in various regimes including online and goal-conditioned RL. In comparison, the proposed method alleviates an introduction of extra techniques other than the shared network architecture, proving superior performance in offline RL benchmarks of diverse environments, e.g. locomotion and manipulation tasks.
>
> Dreamer [5] has brought a notable advancement in model-based RL. Dreamer suggests a world model for decision-making with a considerate design of the latent transition model and reconstructive objective. Since jointly learning an accurate world model and actor in a multi-task environment is challenging, the expensive cost of collecting samples often becomes problematic. In contrast, the proposed method does not necessitate extra modifications of conventional offline RL and proves its sufficient performance gains in comprehensive experiments. Considering previous improvements in representation learning usually involve state-of-the-art design choices (e.g. data augmentation), this paper would contribute to reasonable architectural achievements for researchers by presenting a minimal training structure with verified performance profit.
>
> > The analysis of the effect of predictive pre-training is somewhat unclear to me.
>
> According to [5], temporal difference (TD) learning with a linear function approximator does not guarantee the convergence to true Q function $Q^\pi$. For instance, if $Q^\pi$ stays outside of the column space of the feature matrix, i.e. $Q^\pi\notin C(\Phi)$, TD learning does not converge to the solution of the Bellman equation, $\Phi \theta = T(\Phi\theta)$, because the Bellman equation does not admit a solution inside $C(\Phi)$. Figure 2 illustrates the condition that $Q^\pi$ stays outside the column space of the feature matrix $H_\varphi$. Instead, TD-learning will converge to the minimizer of the mean-squared Bellman error. It is also known that the minimizer of the mean-squared Bellman error corresponds to the solution of the so-called projected Bellman equation, $\Phi\theta = \Pi T(\Phi \theta)$ which admits a unique solution. Mathematically, if the column space of $\Phi$ has full rank, the column space $C(\Phi)$ covers the entire vector space, and this means that $Q^\pi \in C(\Phi)$ and there exists a $\theta^*$ such that $\Phi\theta^* = T(\Phi \theta^*)$. Therefore, TD-learning converges to $\theta^*$. As a result, the right-hand side of the equation (5), the infinity norm of the error between $Q^\pi = \Phi \theta^*$ and $\Pi Q^\pi$, becomes zero. If the feature matrix is singular and $Q^*\notin C(\Phi)$, then there exists a $\theta^*$ such that $\Phi\theta^* =\Pi T(\Phi \theta^*)$. Moreover, the distance between $\Phi\theta^*$ and $T(\Phi \theta^*)$, and the distance between $Q^\pi$ and $\Pi Q^\pi$ decreases when the rank of the column space increases. As a motivating example, the distance between a point and a column space decreases when the column space expands its dimension from 1D (line) to 2D (plane) in general. Based on the reviewer's comment, we will add the related discussions in the revised version.

---

> > ### Author Response · Authors · 2024-11-19
> > **Official Comments by Authors (Cont.)**
> >
> > # Questions
> >
> > > Is the encoder fixed after the pre-training or not? Which one is better?
> >
> > Since freezing the visual feature extractor after pretraining often affects the RL agent's performance [6], we also have considered fixing the parameters of the shared network through an experiment in Appendix G. Surprisingly, TD3+BC trained with the fixed encoder (the shared network) while updating only the last linear layer during remaining RL training outperforms the vanilla TD3+BC in some cases. Based on the results, we speculate that the shared network pretrained with our method would be initialized with a sufficiently better representation for downstream RL training than a randomly initialized network.
> >
> > > Could you explain more about the percentage ratio/percentage rate (line 362)?
> >
> > We appreciate the reviewer's indication about the confusing definition of the pretraining ratio. The "pretraining ratio" stands for the "ratio of the total training *step-time* for pretraining". To avoid ambiguity, we separate the ratio into two different rates in this paper: the pretraining *time-step* ratio and *data-size* ratio. The pretraining *time-step* ratio refers to the ratio of the **pretraining** gradient steps over the **RL training** gradient steps. In contrast, the *data-size* ratio corresponds to the ratio of the **consuming** dataset size over the **original** dataset size where pretraining and RL training **both** share the same dataset. For instance, the pretraining *time-step* ratio of a training setup, where 0.1M pretraining gradient steps and 0.9M RL training gradient steps are given, is 10%. Subsequently, the *data-size* ratio of another training scenario, where only 1M of the total 100M samples are used for both pretraining and RL training, is 1%. In Figure 4, we consider 100% of the *data-size* ratio with 3%, 10%, and 30% of the pretraining *time-step* ratio, respectively (i.e. *0.03M, 0.1M, and 0.3M over the total 1M pretraining gradient steps with the whole dataset.*). In comparison, we choose 1% and 10% of the *data-size* ratio while fixing the pretraining *time-step* ratio as 10%, i.e. 0.1M of 1M samples, from Figure 6 to 9. We have revised the paper to reflect the definition of each pretraining ratio regarding figures and paragraphs.
> >
> > ## Reference
> > [1] M. Schwarzer, A. Anand, R. Goel, R. D. Hjelm, A. Courville, and P. Bachman. Data-efficient reinforcement learning with
> > self-predictive representations. *arXiv preprint arXiv:2007.05929*, 2020.
> >
> > [2] M. Schwarzer, N. Rajkumar, M. Noukhovitch, A. Anand, L. Charlin, R. D. Hjelm, P. Bachman, and A. C. Courville. Pre-
> > training representations for data-efficient reinforcement learning. *Advances in Neural Information Processing Systems*,
> > 34:12686–12699, 2021.
> >
> > [3] D. Yarats, I. Kostrikov, and R. Fergus. Image augmentation is all you need: Regularizing deep reinforcement learning
> > from pixels. In *International conference on learning representations*, 2021.
> >
> > [4] K. He, H. Fan, Y. Wu, S. Xie, and R. Girshick. Momentum contrast for unsupervised visual representation learning. In
> > *Proceedings of the IEEE/CVF conference on computer vision and pattern recognition*, pages 9729–9738, 2020.
> >
> > [5] D. Hafner, J. Pasukonis, J. Ba, and T. Lillicrap. Mastering diverse domains through world models. *arXiv preprint
> > arXiv:2301.04104*, 2023.
> >
> > [6] Yuan, Z., Xue, Z., Yuan, B., Wang, X., Wu, Y., Gao, Y., & Xu, H. Pre-trained image encoder for generalizable visual reinforcement learning. *Advances in Neural Information Processing Systems*, 35, 13022-13037, 2022.

---

> > > ### Comment · Reviewer_DMxg · 2024-11-25
> > >
> > > Thank you for your detailed response. However, I disagree with treating data augmentation or the use of a target encoder as "advanced" training techniques. As a result, I remain unconvinced that the proposed method offers significant advantages over existing approaches of learning predictive representations for sample-efficient RL. Given that the response to the first weakness does not address my concerns sufficiently, I will maintain my original score.

---

> > > > ### Author Response · Authors · 2024-11-25
> > > >
> > > > We thank the reviewer for providing a response to our rebuttal. Before we jump into more comprehensive refutations, we first compare the denoted methods directly with our method.
> > > >
> > > > To begin with, SPR involves representation learning during RL training while our method separates the original training into pretraining and RL training. SPR trains the online encoder by forcing consistent representations over the horizon where the transition model recurrently simulates the future latent representations. Meanwhile, our method trains the shared network by approximating the one-step transition model without recursive predictions. As reported from the paper of SPR, using the target encoder whose parameters are an exponential moving average of the online encoder and projections heads for computing cosine similarity affects the performance of RL significantly. In contrast, our method does not require additional parameters or architectures other than the output size of the shared network with consistently improved performance over baselines. Last but not least, SPR designs the transition model as CNN and evaluates the performance in discrete action problems (Atari 100k) while we propose an MLP-based pretraining methodology applicable to any deep RL and report empirical performance in the continuous control benchmark.
> > > >
> > > > We denoted the target encoder and data augmentation as “advanced training techniques” not to emphasize such techniques were state-of-the-art components in prior works, but to mention that those techniques were dependent on the problem to solve. Considering the sample efficiency problem arose from the difficulty of jointly learning the representation and value [1] and the problem formulation usually contains high-dimensional image input, those methods are 'originally' designed and expected to perform well in such conditions. We maintain that solving the data efficiency problem in the offline RL domain is as important as the sample efficiency problem in previous works and our method demonstrates strong efficiency over several empirical results. We believe that our method would promote data-efficient offline RL for RL practitioners and bring their attention to the data efficiency problem to enable scalable offline RL.
> > > >
> > > > [1] Stooke, A., Lee, K., Abbeel, P., & Laskin, M. (2021, July). Decoupling representation learning from reinforcement learning. In International conference on machine learning (pp. 9870-9879). PMLR.

---

### Official Review · Reviewer_my5h · 2024-10-29

**Soundness:** 2
**Presentation:** 3
**Contribution:** 2
**Rating:** 5
**Confidence:** 4

**Summary:**

The paper proposes using a dynamic loss to pre-train Q-networks, aiming to improve sample efficiency and performance in offline reinforcement learning. A large number of experiments have proved the effectiveness of the algorithm, regardless of the data collection strategies.

**Strengths:**

1. The paper structure is clear and easy to follow.
2. The proposed algorithm is simple and effective, although this simplicity raises some concerns (see weakness 1).
3. The experimental evaluation is comprehensive.

**Weaknesses:**

1. The proposed method appears overly simplistic and lacks novelty. Similar architectures, such as TD-MPC series [1,2] and JOWA [3], also use a shared backbone network to train both dynamics models and value functions, and the latter also pre-trains the backbone using dynamic loss for initialization. The analysis section is largely qualitative while occupies a disproportionate amount of space. Additionally, the relationship between rank and matrix infinity norm is unclear, as is the claim that higher rank is more likely to reduce error (line 257).

2. The absence of comparisons with offline model-based RL algorithms is a significant oversight. Given that the proposed method uses a pre-training approach similar to model-based methods and offline model-based RL also aims to improve data-efficiency, comparisons with algorithms such as MoRel [4], MOPO [5], COMBO [6], and RAMBO [7] would be highly relevant. I recommend including at least two offline model-based RL algorithms in the comparisons to ensure fairness and comprehensiveness.

3. Directly extracting AWAC and CQL algorithm results from the TD3+BC paper in Table 2 is inappropriate. I suggest either using results from the original papers of these algorithms or, preferably, reproducing the experiments. The reported results for AWAC and CQL appear to be significantly lower than those from a reputable offline RL algorithm library [8].

4. Obviously, the authors are not the first to raise the problem of data efficiency in the offline RL field (line 501). In addition to offline model-based RL, many other works [9,10,11] have also studied the problem of data efficiency, experimenting on datasets with different downsampling rates.

[1] Hansen N, Wang X, Su H. Temporal difference learning for model predictive control[J]. arXiv preprint arXiv:2203.04955, 2022.

[2] Hansen N, Su H, Wang X. Td-mpc2: Scalable, robust world models for continuous control[J]. arXiv preprint arXiv:2310.16828, 2023.

[3] Cheng J, Qiao R, Xiong G, et al. Scaling Offline Model-Based RL via Jointly-Optimized World-Action Model Pretraining[J]. arXiv preprint arXiv:2410.00564, 2024.

[4] Kidambi R, Rajeswaran A, Netrapalli P, et al. Morel: Model-based offline reinforcement learning[J]. Advances in neural information processing systems, 2020, 33: 21810-21823.

[5] Yu T, Thomas G, Yu L, et al. Mopo: Model-based offline policy optimization[J]. Advances in Neural Information Processing Systems, 2020, 33: 14129-14142.

[6] Yu T, Kumar A, Rafailov R, et al. Combo: Conservative offline model-based policy optimization[J]. Advances in neural information processing systems, 2021, 34: 28954-28967.

[7] Rigter M, Lacerda B, Hawes N. Rambo-rl: Robust adversarial model-based offline reinforcement learning[J]. Advances in neural information processing systems, 2022, 35: 16082-16097.

[8] https://github.com/tinkoff-ai/CORL

[9] Agarwal R, Schuurmans D, Norouzi M. An optimistic perspective on offline reinforcement learning[C]//International conference on machine learning. PMLR, 2020: 104-114.

[10] Kumar A, Zhou A, Tucker G, et al. Conservative q-learning for offline reinforcement learning[J]. Advances in Neural Information Processing Systems, 2020, 33: 1179-1191.

[11] Kumar A, Agarwal R, Ghosh D, et al. Implicit Under-Parameterization Inhibits Data-Efficient Deep Reinforcement Learning[C]. International Conference on Learning Representations, 2020.

**Questions:**

1. The paper mentions "supervised learning domain (Chebotar et al., 2023 ...)" in reference to Q-former. However, doesn't Q-former use TD-learning, which falls under the RL domain rather than supervised learning?

2. Could you clarify what "pretraining ratios" (line 362) refers to? Is it the ratio of pre-training steps to total training steps?

3. In some cases (e.g., Figure 8 SMM), increasing the data amount leads to a decrease in offline RL performance. Could you provide a qualitative explanation for this phenomenon?

---

> ### Author Response · Authors · 2024-11-19
>
> We appreciate the reviewer for providing meticulous review and comments. With the reviewer's commitment, we improve the overall quality of this paper by a significant margin. We hope that our comments clearly answer the reviewer's questions and settle the insufficiency.
>
> # Weaknesses
>
> > Similar architectures, such as TD-MPC series [1,2] and JOWA [3], also use a shared backbone network to train both dynamics models and value functions, and the latter also pre-trains the backbone using dynamic loss for initialization.
>
> TDMPC and TDMPC2 are model-based single and multi-task RL approaches, which recursively feed the output of the same network (i.e. the encoder and task embedding network) for the transition model and value learning. The outputs of the shared backbone networks correspond to the latent representation and task embedding vector, respectively, and most latent model-based RL approaches including TDMPC reuse the outputs for the transition model and value learning. On the other hand, the proposed method presents a shared network architecture resembling the dueling architecture [4] to pretrain the shared backbone network with a separated stream (a header) of the transition model and Q-network. Additionally, this paper presents a two-phase training scheme: the transition model combined with the shared network is trained with the transition dynamics prediction task in the first phase and the Q-network, consisting of an MLP header and the shared network initialized with the parameter of the shared network in the first phase, is trained with the downstream RL value learning task in the second phase.
>
> JOWA is an offline world model for multi-task RL with a shared Transformer backbone network for sequential a next-token prediction task. By modeling the decision-making problem to the sequential token prediction task, the backbone network, tokenizer, and header are trained in a supervised manner with the offline dataset. While the main purpose of JOWA is scaling an offline world model across multiple tasks with generalized performance over unseen tasks, this paper intends to improve the data efficiency of conventional offline RL approaches in single-task RL. Furthermore, the proposed method alleviates additional training after offline RL training with a novel two-phase training strategy while JOWA allows few-shot fine-tuning for sample efficient transfer with a multi-game environment. Even with a similar purpose of data efficiency, the proposed method entails a minimal algorithmic change with a consistent training budget compared to previous approaches. Based on the reviewer's comment, we will add the related discussions in the revised version.
>
> > The relationship between rank and matrix infinity norm is unclear, as is the claim that higher rank is more likely to reduce error.
>
> According to [5], temporal difference (TD) learning with a linear function approximator does not guarantee the convergence to true Q function $Q^\pi$. For instance, if $Q^\pi$ stays outside of the column space of the feature matrix, i.e. $Q^\pi\notin C(\Phi)$, TD learning does not converge to the solution of the Bellman equation, $\Phi \theta = T(\Phi\theta)$, because the Bellman equation does not admit a solution inside $C(\Phi)$. Figure 2 illustrates the condition that $Q^\pi$ stays outside the column space of the feature matrix $H_\varphi$. Instead, TD-learning will converge to the minimizer of the mean-squared Bellman error. It is also known that the minimizer of the mean-squared Bellman error corresponds to the solution of the so-called projected Bellman equation, $\Phi\theta = \Pi T(\Phi \theta)$ which admits a unique solution. Mathematically, if the column space of $\Phi$ has full rank, the column space $C(\Phi)$ covers the entire vector space, and this means that $Q^\pi \in C(\Phi)$ and there exists a $\theta^*$ such that $\Phi\theta^* = T(\Phi \theta^*)$. Therefore, TD-learning converges to $\theta^*$. As a result, the right-hand side of the equation (5), the infinity norm of the error between $Q^\pi = \Phi \theta^*$ and $\Pi Q^\pi$, becomes zero. If the feature matrix is singular and $Q^*\notin C(\Phi)$, then there exists a $\theta^*$ such that $\Phi\theta^* =\Pi T(\Phi \theta^*)$. Moreover, the distance between $\Phi\theta^*$ and $T(\Phi \theta^*)$, and the distance between $Q^\pi$ and $\Pi Q^\pi$ decreases when the rank of the column space increases. As a motivating example, the distance between a point and a column space decreases when the column space expands its dimension from 1D (line) to 2D (plane) in general. Based on the reviewer's comment, we will add the related discussions in the revised version.

---

> > ### Author Response · Authors · 2024-11-19
> > **Official Comments by Authors (Cont.)**
> >
> > > The absence of comparisons with offline model-based RL algorithms is a significant oversight.
> >
> > We thank the reviewer for suggesting remarkable opinions concerning extra experiments. To improve the completeness of this paper, we have conducted additional experiments and reflected the results in Figure 7 of the revised paper. We consider two offline model-based methods, MOPO [6] and MOBILE [7], as representatives in the experiment. As a result, popular offline model-free RL combined with the proposed method manifests superior or similar performance comparing offline model-based RL. While the offline model-based RL approaches spend far more than 1M gradient steps for the model and RL training in general, the proposed method requires 1M gradient steps for both pretraining and original RL training. Even with lower training costs, the proposed method outperforms the model-based approaches in most cases in Figure 7, indicating that our method exhibits better training efficiency and performance than offline model-based RL approaches.
> >
> > > Directly extracting AWAC and CQL algorithm results from the TD3+BC paper in Table 2 is inappropriate.
> >
> > In the original manuscript, we have referred [8] for baseline scores of AWAC [9], CQL [10], and TD3+BC [8] in our works. Actually, [8] also refers to each original paper for the scores of AWAC and CQL, comparing the performance of TD3+BC with them. Therefore, the scores are the same. However, we admit that it might be confusing for authors to compare the results. To clarify the source of each reported score, we revise the descriptions of the baselines in the modified paper (in Section 5).
> >
> > > The authors are not the first to raise the problem of data efficiency in the offline RL field (line 501).
> >
> > We apologize for denoting an inappropriate description of the contribution of this paper. Therefore, we remove the expression and will reinforce our argument properly by suggesting a few refuting viewpoints. Most of all, the offline model-based approaches, including [6, 7, 11, 12, 13], could resolve the data efficiency problem by generating in-distribution samples. However, experimental results of previous offline model-based RL have omitted additional data efficiency conditions, e.g. a performance evaluation when the dataset size is reduced. As the revised paper suggests, supplementary experiments (Figure 7) related to similar baselines indicate that the proposed method proves superior data efficiency over the baselines. Additionally, other branches (e.g. [10, 14, 15]) have reported the experiments for reduced datasets in the discrete action space environment whereas this paper considers the continuous action space under diverse environments. In this work, we present our method as effective under the offline RL scheme regardless of the data quality of the behavior (Section 5.2) and the shape of the data distribution (Section 5.3) through extensive experiments. We believe that our effort to aggregate the performance of popular offline RL algorithms in considerable data efficiency evaluation settings can be a main contribution.

---

> > > ### Author Response · Authors · 2024-11-19
> > > **Official Comments by Authors (Cont.)**
> > >
> > > # Questions
> > >
> > > > Doesn't Q-former use TD-learning, which falls under the RL domain rather than supervised learning?
> > >
> > > We have referred to [16] as a recent study in offline RL that follows the dominant paradigm of the supervised learning domain: focusing on a large dataset and a scalable neural network model. As the reviewer points out, [16] proposes a novel architecture combining Transformer with CQL [10] for scalable generalist policy learning, which involves TD learning with sequential next token prediction. To clarify our intention, we remove the confusing sentence "following the paradigm of the supervised learning domain" in the introduction paragraph and move the reference to the offline RL section in the revision.
> > >
> > > > Could you clarify what "pretraining ratios" (line 362) refers to?
> > >
> > > As the reviewer conjectures, the "pretraining ratio" corresponds to "the ratio of pre-training steps to total training steps". We apologize to the reviewer for being confused with the definition of the pretraining ratio. To avoid ambiguity, we separate the ratio into two different rates in this paper: the pretraining *time-step* ratio and *data-size* ratio. The pretraining *time-step* ratio refers to the ratio of the **pretraining** gradient steps over the **RL training** gradient steps. In contrast, the *data-size* ratio corresponds to the ratio of the **consuming** dataset size over the **original** dataset size where pretraining and RL training **both** share the same dataset. For instance, the pretraining *time-step* ratio of a training setup, where 0.1M pretraining gradient steps and 0.9M RL training gradient steps are given, is 10%. Subsequently, the *data-size* ratio of another training scenario, where only 1M of the total 100M samples are used for both pretraining and RL training, is 1%. In Figure 4, we consider 100% of the *data-size* ratio with 3%, 10%, and 30% of the pretraining *time-step* ratio, respectively (i.e. *0.03M, 0.1M, and 0.3M over the total 1M pretraining gradient steps with the whole dataset.*). In comparison, we choose 1% and 10% of the *data-size* ratio while fixing the pretraining *time-step* ratio as 10%, i.e. 0.1M of 1M samples, from Figure 6 to 9. We have revised the paper to reflect the definition of each pretraining ratio in regarding figures and paragraphs.
> > >
> > > > In some cases (e.g., Figure 8 SMM), increasing the data amount leads to a decrease in offline RL performance.
> > >
> > > We thank the reviewer for raising a valid question related to the empirical result. We have conducted additional experiments to justify the unusual situation with some ablation studies (e.g. random seeds). We suggest a new result in our revised paper (Figure 8) by fixing the same dataset collected with SMM and modulating the randomness of RL training.

---

> > > > ### Author Response · Authors · 2024-11-19
> > > > **Reference**
> > > >
> > > > [1] N. Hansen, X. Wang, and H. Su. Temporal difference learning for model predictive control. (arXiv:2203.04955), July 2022.
> > > > doi: 10.48550/arXiv.2203.04955. URL http://arxiv.org/abs/2203.04955. arXiv:2203.04955 [cs].
> > > >
> > > > [2] N. Hansen, H. Su, and X. Wang. Td-mpc2: Scalable, robust world models for continuous control. *arXiv preprint
> > > > arXiv:2310.16828*, 2023.
> > > >
> > > > [3] J. Cheng, R. Qiao, G. Xiong, Q. Miao, Y. Ma, B. Li, Y. Li, and Y. Lv. Scaling offline model-based rl via jointly-optimized
> > > > world-action model pretraining. *arXiv preprint arXiv:2410.00564*, 2024.
> > > >
> > > > [4] Z. Wang, T. Schaul, M. Hessel, H. Hasselt, M. Lanctot, and N. Freitas. Dueling network architectures for deep reinforce-
> > > > ment learning. In *International conference on machine learning*, pages 1995–2003. PMLR, 2016.
> > > >
> > > > [5] H. R. Maei. Gradient temporal-difference learning algorithms. 2011.
> > > >
> > > > [6] T. Yu, G. Thomas, L. Yu, S. Ermon, J. Y. Zou, S. Levine, C. Finn, and T. Ma. Mopo: Model-based offline policy
> > > > optimization. *Advances in Neural Information Processing Systems*, 33:14129–14142, 2020.
> > > >
> > > > [7] Y. Sun, J. Zhang, C. Jia, H. Lin, J. Ye, and Y. Yu. Model-bellman inconsistency for model-based offline reinforcement
> > > > learning. In *International Conference on Machine Learning*, pages 33177–33194. PMLR, 2023.
> > > >
> > > > [8] S. Fujimoto and S. S. Gu. A minimalist approach to offline reinforcement learning. *Advances in neural information
> > > > processing systems*, 34:20132–20145, 2021.
> > > >
> > > > [9] A. Nair, A. Gupta, M. Dalal, and S. Levine. Awac: Accelerating online reinforcement learning with offline datasets. *arXiv
> > > > preprint arXiv:2006.09359*, 2020.
> > > >
> > > > [10] A. Kumar, A. Zhou, G. Tucker, and S. Levine. Conservative q-learning for offline reinforcement learning. *Advances in
> > > > Neural Information Processing Systems*, 33:1179–1191, 2020b.
> > > >
> > > > [11] R. Kidambi, A. Rajeswaran, P. Netrapalli, and T. Joachims. Morel: Model-based offline reinforcement learning. *Advances
> > > > in neural information processing systems*, 33:21810–21823, 2020.
> > > >
> > > > [12] T. Yu, A. Kumar, R. Rafailov, A. Rajeswaran, S. Levine, and C. Finn. Combo: Conservative offline model-based policy
> > > > optimization. *Advances in neural information processing systems*, 34:28954–28967, 2021.
> > > >
> > > > [13] M. Rigter, B. Lacerda, and N. Hawes. Rambo-rl: Robust adversarial model-based offline reinforcement learning. *Advances
> > > > in neural information processing systems*, 35:16082–16097, 2022.
> > > >
> > > > [14] R. Agarwal, D. Schuurmans, and M. Norouzi. An optimistic perspective on offline reinforcement learning. In *International
> > > > conference on machine learning*, pages 104–114. PMLR, 2020.
> > > >
> > > > [15] A. Kumar, R. Agarwal, D. Ghosh, and S. Levine. Implicit under-parameterization inhibits data-efficient deep reinforcement
> > > > learning. *arXiv preprint arXiv:2010.14498*, 2020a.
> > > >
> > > > [16] Y. Chebotar, Q. Vuong, K. Hausman, F. Xia, Y. Lu, A. Irpan, A. Kumar, T. Yu, A. Herzog, K. Pertsch, et al. Q-
> > > > transformer: Scalable offline reinforcement learning via autoregressive q-functions. In *Conference on Robot Learning*,
> > > > pages 3909–3928. PMLR, 2023.

---

> ### Comment · Reviewer_my5h · 2024-11-25
>
> Thank you for your detailed response. The explanation of the relationship between rank and error helps me understand your analysis in the manuscript. The additional experiments of offline model-based RL algorithms greatly improves the completeness of the paper and supports the contribution on data efficiency. The discussions and experiments address most of my concerns.
>
> However, I still have the main concern on the novelty. Your first response describes the differences between your method and TD-MPC series and JOWA. But these differences seem minor and not critical enough to indicate that what they can't or have difficulty doing, your method does.
>
> Moreover, the issue that increasing the data amount leads to a decrease in offline RL performance exists in many results of the manuscript (e.g., Mobile in Figure7, hopper with 10% and 100% data amount; CQL+ours in Figure7, walker2d with 10% and 100% data amount). In fact, based on my experience and knowledge, I know this is common in offline RL, i.e., it's not necessarily the case that increasing the data will produce better results. I'd just like to hear the author's opinion on this issue, do you think it's due to randomness (i.e. random seeds) or an algorithmic issue, or is it inherent to offline RL problem.

---

> > ### Author Response · Authors · 2024-11-25
> >
> > We appreciate that our explanations can resolve most of the reviewer's concerns. We are aware of the reviewer's contribution to enhancing the overall quality of this paper and are pleased to discuss further questions.
> >
> > > Differences with prior model-based approaches seem minor and not critical enough...
> >
> > In addition to the first response, we consider the main contribution of this paper as suggesting a different viewpoint of how the transition model is used and proposing a more cost-efficient method. Since most model-based RL methods including TD-MPC involve an iterative process of model learning and planning with simulation, the model-based RL approach can achieve superior sample efficiency by generating synthetic future trajectories. While the increased network size of the model enables scalable multi-task RL (e.g., TD-MPC2), most model-based RL methods suffer from poor wall-clock time efficiency due to the increased amount of network forward and inverse passes compared to model-free RL [1]. We propose a different view of exploiting the transition model learning for data-efficient offline RL by the shared network architecture, which requires a minimal change of the underlying offline RL backbone. Although we do not report the empirical comparison of wall-clock time efficiency since the results can vary on the local environment (e.g., computing machine) a lot, we find that our method with the model-free method exhibits superior efficiency about three times better compared to model-based approaches (e.g., MOPO in Figure 7).
> >
> > Since learning a generalizable offline world model across multiple tasks usually requires prohibitively large samples, reducing the number of samples can become critical in such problem formulation. Furthermore, JOWA's main focus relies on how to scale TD-based offline RL for simultaneous general-purpose representation and decision-making with the Transformer architecture. In contrast, we focus on the 'general' offline RL formulation to enable data-efficient offline RL and stress the importance of data efficiency in offline RL by suggesting extensive empirical results of the plug-and-play method. We believe that our method would boost further deployments of offline RL to more realistic applications where collecting a large amount of offline data is limited (e.g., robotics).
> >
> > > Increasing the amount of offline data does not always lead to an increase in the performance of offline RL
> >
> > We have discussed this phenomenon with the authors and concluded that there might be two possible reasons: randomness or problem formulation. First, as we describe in our first response, randomness including the seed could affect the consistency of the offline RL methods' performance. Considering that our method is the pretraining method initializing the partial networks of the Q-network and there have been extensive lines of research in the deep learning field to find a better network initialization method [2, 3], the randomly initialized parameters that are not appropriate for downstream RL optimization might be attributed to the unstable performance of deep RL.
> >
> > Another potential explanation is the problem formulation in offline RL. Following the conventional problem formulation of offline RL, most previous approaches including state-of-the-art methods (e.g., CQL or IQL) mainly consider the problem of alleviating over-optimistic Q value estimation when out-of-distributional input pairs consisting of state and action are given during online evaluation. By pushing boundaries in prior literature, current popular offline RL methods can handle the problem effectively and prove superior performance in diverse domains to some extent. However, most problems we are interested in become more complex and diverse, leading to limited results in prior literature where the offline data might lack enough coverage of the entire joint state and action space to alleviate the traditional offline RL problem. Though several works have considered a novel offline data collection setting like in [4],  we believe that our work can deliver a message that further investigations concerning the data itself should be conducted more in the offline RL field.
> >
> > [1] Landolfi, N. C., Thomas, G., & Ma, T. (2019). A model-based approach for sample-efficient multi-task reinforcement learning. arXiv preprint arXiv:1907.04964.
> >
> > [2] He, K., Zhang, X., Ren, S., & Sun, J. (2015). Delving deep into rectifiers: Surpassing human-level performance on imagenet classification. In Proceedings of the IEEE international conference on computer vision (pp. 1026-1034).
> >
> > [3] Hu, W., Xiao, L., & Pennington, J. (2020). Provable benefit of orthogonal initialization in optimizing deep linear networks. arXiv preprint arXiv:2001.05992.
> >
> > [4] Yarats, D., Brandfonbrener, D., Liu, H., Laskin, M., Abbeel, P., Lazaric, A., & Pinto, L. (2022). Don't change the algorithm, change the data: Exploratory data for offline reinforcement learning. arXiv preprint arXiv:2201.13425.

---

> > > ### Comment · Reviewer_my5h · 2024-11-26
> > >
> > > Thank you for your detailed response again. Although your response about novelty still can't convince me of the critical differences between methods, considering that this paper did conduct detailed experiments to verify the effectiveness and improvement on data-efficiency of the method, I decide to increase the rating to 5.

---

### Official Review · Reviewer_bsTd · 2024-11-03

**Soundness:** 3
**Presentation:** 2
**Contribution:** 3
**Rating:** 6
**Confidence:** 3

**Summary:**

In this paper, the authors focus on the problem of data-efficient offline reinforcement learning. To this end, they propose a very simple approach to improve downstream performance via pre-training. They utilize a Q-network with shared weights in addition to a future state prediction task on offline data. They provide mathematical justification for their approach and then evaluate their approach extensively with a number of other offline baselines on multiple datasets (D4RL, Robomimic). They show a quantitative margin of improvement in most settings.

**Strengths:**

1. The approach is very simple. It's largely just regression on future states.

2. Experimental evaluation is extensive with multiple environments used as well as multiple baselines for ablation.

3. Quantitative results are strong. The margin of performance over baselines shows promise.

**Weaknesses:**

1. This approach seems to be very similar to previous works which forecast future state information. For example, Self Predictive Representations (Schwarzer et al. 2020) and Predictive Belief Representations (Guo et al. 2018) predict future state information to improve performance of RL-trained agents.

2. There are some issues with presentation in the paper. For example, in Table 2, it appears that some of the baseline numbers are omitted (such as Expert IQL).

**Questions:**

1. Could you please elaborate on the novelty of the approach over previous similar work? How is this approach distinct from some of the papers mentioned above?

2. Could you clarify some of the omitted numbers in Table 2?

---

> ### Author Response · Authors · 2024-11-19
>
> We thank the reviewer for raising reasonable and plausible concerns through delicate analysis. In the spirit of openness, we believe successful discussions with the reviewer would contribute to the qualitative revision. Regarding the suggested weaknesses and questions, we hope the explanations below address the informed issues satisfactorily.
>
> # Weaknesses & Questions
>
> > This approach seems to be very similar to previous works which forecast future state information.
>
> While one might indicate that the proposed method lacks novelty, we contend that our method contains a few distinct ideas and purposes. As the reviewer states, the proposed method pretrains the neural networks with the next state prediction objective to improve an underlying RL agent's performance and data efficiency similar to [1, 2]. However, [1] has proposed an online training method in a self-supervised learning manner whereas the proposed method considers supervised learning for pretraining. Since the self-predictive task in [1] is conducted in latent space, representation learning is essentially involved with the task. Therefore, adopting advanced training techniques including data augmentation [3] and the use of a target encoder [4] significantly affect the RL agent's performance. Additionally, [1] suggests a self-supervised representation learning with the latent transition prediction task in the online RL regime. In comparison, the proposed method alleviates an introduction of extra techniques other than the shared network architecture, proving superior performance in offline RL benchmarks of diverse environments, e.g. locomotion and manipulation tasks.
>
> [2] has presented an unsupervised learning method that encodes the *belief state* capturing sufficient information of the hidden true state from a past interaction history. In other words, the main interest of [2] is how the neural network architecture trained with unsupervised learning extracts adequate information concerning the true state in POMDP, not how the underlying RL method given rich representation performs decision-making problem well. Specifically, the network architecture in [2] is based on GRU, RNN based sequential network, and predicts a next observation $o_{t+1}$ using action $a_t$ and a belief state $b_t$ that contains the partial information of the previous trajectory. Conversely, the proposed method in this paper is implemented on MLP with the shared network architecture and predicts the next state $s_{t+1}$ using current state $s_t$ and action $a_t$ without a past history. Finally, this paper demonstrates the proposed method enables data-efficient offline RL with extensive experiments. Based on the reviewer's comment, we will add the related discussions in the revised version.
>
> > There are omitted numbers in Table 2.
>
> In Table 2, we brought the reported scores of original offline RL algorithms from each paper in the D4RL benchmark [5,6,7,8]. D4RL benchmark provides a tool for algorithmic search in offline RL with diverse pre-collected datasets. Most of the previous approaches in offline RL have considered mainly 5 types of offline data in locomotion tasks: *random, medium, medium-replay, medium-expert*, and *expert*. Unfortunately, not all these methods have reported the performance of a novel method in the same experimental configuration. For instance, the IQL [6] paper does not provide the score of experiments trained with the *random* and *expert* datasets.
>
> ## Reference
> [1] M. Schwarzer, A. Anand, R. Goel, R. D. Hjelm, A. Courville, and P. Bachman. Data-efficient reinforcement learning with
> self-predictive representations. *arXiv preprint arXiv:2007.05929*, 2020.
>
> [2] Z. D. Guo, M. G. Azar, B. Piot, B. A. Pires, and R. Munos. Neural predictive belief representations. *arXiv preprint
> arXiv:1811.06407*, 2018.
>
> [3] D. Yarats, I. Kostrikov, and R. Fergus. Image augmentation is all you need: Regularizing deep reinforcement learning
> from pixels. In *International conference on learning representations*, 2021.
>
> [4] K. He, H. Fan, Y. Wu, S. Xie, and R. Girshick. Momentum contrast for unsupervised visual representation learning. In
> *Proceedings of the IEEE/CVF conference on computer vision and pattern recognition*, pages 9729–9738, 2020.
>
> [5] S. Fujimoto and S. S. Gu. A minimalist approach to offline reinforcement learning. *Advances in neural information
> processing systems*, 34:20132–20145, 2021.
>
> [6] I. Kostrikov, A. Nair, and S. Levine. Offline reinforcement learning with implicit q-learning. *arXiv preprint
> arXiv:2110.06169*, 2021.
>
> [7] A. Kumar, A. Zhou, G. Tucker, and S. Levine. Conservative q-learning for offline reinforcement learning. *Advances in
> Neural Information Processing Systems*, 33:1179–1191, 2020b.
>
> [8] A. Nair, A. Gupta, M. Dalal, and S. Levine. Awac: Accelerating online reinforcement learning with offline datasets. *arXiv
> preprint arXiv:2006.09359*, 2020.

---

> > ### Comment · Reviewer_bsTd · 2024-11-26
> >
> > Thank you for your detailed responses to my concerns. I stand by my rating.

---

### Official Review · Reviewer_dfjX · 2024-11-04

**Soundness:** 4
**Presentation:** 2
**Contribution:** 3
**Rating:** 6
**Confidence:** 3

**Summary:**

This paper proposed a pretraining method for Offline RL. The method first pre-trains the Q-function to predict the forward dynamics of task based on a static dataset. The pre-train Q-function is then used as initilzation for standard offline RL training. The authors validate their proposed method across multiple offline RL benchmarks and baselines.

**Strengths:**

- The paper presents extensive experimental results of the proposed methods on both offline and online RL across multiple benchmarks and baselines that validates the effectiveness of the proposed method.
- The presented result is quite flexible and can be easily plugged into most offline/online RL methods.

**Weaknesses:**

- The writing of the paper can be improved. There are some awkwardly written sentences (line 99 "..ability of an offline RL algorithm whether an agent can learn the desired policy...") and inconsistent switching between active and passive voice (line 192 "...underlying insights behind the proposed method are discussed", line 356 "The learning curves of TD3+BD are illustrated in Figure 3...", etc) that makes the paper hard to read.
- In Section 5.1, It not entirely clear to me what the authors mean by "pretraining ratios". Is the ratio of the amount of data used for pretraining or ratio of the total training time for pretraining?

**Questions:**

- The proposed method bares some similarities to Model-based RL methods as the paper proposed to first pretrain on the environments forward dynamics within the Q-function which is similar to how Model-based RL methods learn a seperate model to model the transition dynamics of the environment. Do the authors see any benefit/trade-offs in learning the dynamics within the Q-function itself vs learning the dynamics model seperately?

---

> ### Author Response · Authors · 2024-11-19
>
> We appreciate the reviewer for providing noteworthy feedback regarding our paper. We believe the reviewer's meticulous and concise comments would significantly improve the paper in the qualitative and quantitative context. Concerning suggested weaknesses and questions, we present our explanations in detail as follows.
>
> # Weaknesses
>
> > The writing of the paper can be improved...
>
> We apologize to the reviewer for being uncomfortable with the inconsistent passive and active voice while reviewing the paper. We have revised the paper to deliver our idea more clearly and logically. Please check the modified paper and leave a comment if there are remaining concerns.
>
> > In Section 5.1, It is not entirely clear to me what the authors mean by "pretraining ratios...
>
> To begin with, we apologize to the reviewer for being confused with the definition of the pretraining ratio. As the reviewer denotes, the latter is a more accurate expression: "ratio of the total training time for pretraining". To avoid ambiguity, we separate the ratio into two different rates in this paper: the pretraining *time-step* ratio and *data-size* ratio. The pretraining *time-step* ratio refers to the ratio of the **pretraining** gradient steps over the **RL training** gradient steps. In contrast, the *data-size* ratio corresponds to the ratio of the **consuming** dataset size over the **original** dataset size where pretraining and RL training **both** share the same dataset. For instance, the pretraining *time-step* ratio of a training setup, where 0.1M pretraining gradient steps and 0.9M RL training gradient steps are given, is 10%. Subsequently, the *data-size* ratio of another training scenario, where only 1M of the total 100M samples are used for both pretraining and RL training, is 1%. In Figure 4, we consider 100% of the *data-size* ratio with 3%, 10%, and 30% of the pretraining *time-step* ratio, respectively (i.e. *0.03M, 0.1M, and 0.3M over the total 1M pretraining gradient steps with the whole dataset.*). In comparison, we choose 1% and 10% of the *data-size* ratio while fixing the pretraining *time-step* ratio as 10%, i.e. 0.1M of 1M samples, from Figure 6 to 9. We have revised the paper to reflect the definition of each pretraining ratio in regarding figures and paragraphs.
>
> # Questions
>
> > Do the authors see any benefit/trade-offs in learning the dynamics within the Q-function itself vs learning the dynamics model separately?
>
> Regarding the similarity between our pretraining method and model-based RL, we suggest a few benefits over model-based RL in terms of performance and efficiency. As the reviewer indicates, typical model-based RL trains a separate model that mimics the transition dynamics of the environment. It is well known that model-based RL demonstrates superior sample efficiency by rolling out the learned model to generate additional transition samples and exploiting the samples for downstream RL training. However, decreased prediction capability under the mismatch between the learned and true model is often attributed to a possible drawback of model-based RL. In addition, learning an accurate model under the offline RL scheme, where the unseen state and action pairs significantly decrease the prediction ability of the learned networks, produces a unique challenge that might excel when available data is small. However, our method alleviates the potential downsides of model-based RL by only pretraining the shared network architecture with the same transition dynamics prediction task and continuing downstream RL training with the pretrained shared model. We contend that our method manifests a lower training cost than the cost of offline model-based RL in Figure 7 of the revised paper. While the offline model-based RL approaches spend far more than 1M gradient steps for the model and RL training in general, our method requires 1M gradient steps for both pretraining and original RL training. Even with lower training costs, our method outperforms the model-based approaches in Figure 7, indicating that our method exhibits better training efficiency and performance than learning the separate dynamics model. Based on the reviewer's comment, we have appended the related discussions in the revised version.
>
> Again, we thank the reviewer for providing delicate and conspicuous feedback to this paper. Based on the comments, we have improved the overall quality and legibility of the paper.

---

> > ### Comment · Reviewer_dfjX · 2024-11-26
> >
> > Thanks for the response. I have no further questions.

---

### Meta-Review · Area_Chair_2LWt · 2024-12-20

**Metareview:**

Summary of the paper:

This paper tackles the challenge of data-efficient offline reinforcement learning (RL) by introducing a pretraining method for Q-functions. The core idea is to pretrain the Q-function to predict forward dynamics of tasks using a static dataset, followed by using the pretrained Q-function as initialization for standard offline RL training. A key feature of the proposed approach is the incorporation of a Q-network with shared weights alongside a future state prediction task during pretraining, complemented by a dynamic loss function to improve sample efficiency and performance. Extensive evaluations across multiple offline RL benchmarks and datasets, including D4RL and Robomimic, highlight the effectiveness of the method, demonstrating consistent improvements over various baselines regardless of data collection strategies.

Strengths: The reviewers acknowledge several key strengths in the paper: 1) addressing an important challenge of data-efficient offline RL, 2) proposing a simple yet effective pretraining method for Q-functions, 3) providing interesting theoretical analysis to explain the benefits of the predictive pretraining phase, and 4) conducting extensive experimental evaluations across multiple benchmarks with strong quantitative results and informative ablations.

Weaknesses and missing in the submission: While the rebuttal has effectively addressed some of the reviewers' concerns, there remains a consensus among reviewers regarding the weaknesses and limitations of the paper, as outlined below:

1) The primary weakness identified is the lack of technical novelty. The proposed method closely resembles prior works, such as Self Predictive Representations (Schwarzer et al., 2020), Predictive Belief Representations (Guo et al., 2018), and methods like TD-MPC and JOWA, which also leverage shared backbone architectures and predictive pretraining. The distinctions between the proposed method and these prior works are perceived as minor and insufficiently critical to establish significant novelty. Additionally, concerns were raised about the method’s simplicity, with reviewers questioning whether it provides substantial advantages over existing approaches. Despite extensive discussions between the authors and reviewers, concerns regarding the novelty of the work remained unresolved, particularly for Reviewers my5h and DMxg.

2)  Limited theoretical rigor, with some claims (e.g., the relationship between rank, matrix infinity norm, and error) lacking clear explanations or proofs. The theoretical analysis is largely qualitative, and a more coherent and detailed explanation of key implications (e.g., how predictive pretraining improves downstream RL) is necessary.

3) Lack of comparisons with relevant offline model-based RL methods (e.g., MoRel, MOPO, COMBO, RAMBO), which is a significant oversight given the method’s conceptual similarities to model-based approaches. While some comparisons were provided during the rebuttal phase, they were not sufficiently thorough. Incorporating more detailed and rigorous comparisons would significantly improve the fairness and comprehensiveness of the evaluation.

4) The issue of decreasing performance with increasing data (observed in some experiments) remains inadequately explained.

5) Reviewers have expressed concerns about the paper's presentation, noting issues such as inconsistent writing style, unclear terminology (e.g., "pretraining ratios"), and missing details in tables (e.g., omitted baseline numbers in Table 2). These issues detract from the paper’s readability and accessibility.

**Additional Comments On Reviewer Discussion:**

Despite the authors’ efforts to improve the submission and provide clarifications during the discussion phase, the reviewers remained unconvinced, maintaining a borderline assessment.  By the end of the discussion stage, all four reviewers kept their ratings as marginally above or below the acceptance threshold.

In light of the reviewers’ comments and recommendations, the area chairs concluded that the weaknesses outweigh the strengths of the paper. The area chairs acknowledge the paper's potential and its implications for practical applications. Addressing the remaining concerns – such as demonstrating stronger technical novelty, providing thorough comparisons, offering a deeper discussion of the results, and undergoing substantial revision – could significantly enhance the paper in a future submission cycle.

---

### Decision · Program_Chairs · 2025-01-22

Reject